# Near-lifespan longitudinal tracking of brain microvascular morphology, topology, and flow in male mice

Konrad W. Walek[1,10], Sabina Stefan[2,10], Jang-Hoon Lee[2], Pooja Puttigampala[3], Anna H. Kim[4], Seong Wook Park[4], Paul J. Marchand [5], Frederic Lesage[5], Tao Liu[6], Yu-Wen Alvin Huang [1,7,8], David A. Boas [9], Christopher Moore [4,8] & Jonghwan Lee [2,8] ✉

In age-related neurodegenerative diseases, pathology often develops slowly across the lifespan. As one example, in diseases such as Alzheimer's, vascular decline is believed to onset decades ahead of symptomology. However, challenges inherent in current microscopic methods make longitudinal tracking of such vascular decline difficult. Here, we describe a suite of methods for measuring brain vascular dynamics and anatomy in mice for over seven months in the same field of view. This approach is enabled by advances in optical coherence tomography (OCT) and image processing algorithms including deep learning. These integrated methods enabled us to simultaneously monitor distinct vascular properties spanning morphology, topology, and function of the microvasculature across all scales: large pial vessels, penetrating cortical vessels, and capillaries. We have demonstrated this technical capability in wild-type and 3xTg male mice. The capability will allow comprehensive and longitudinal study of a broad range of progressive vascular diseases, and normal aging, in key model systems.

There is an increasing need for methods to longitudinally track microvascular alterations in the aging brain. As one example, in Alzheimer's disease, vascular decline is thought to emerge up to decades before the onset of cognitive symptoms and is widely viewed as one potential contributor to symptomology[1]. Similarly, changes in vascular structure and dynamics are a signature of several other progressive neurodegenerative diseases, including Parkinson disease[2], Huntington disease[3], and multiple sclerosis[4], suggesting a direct role in disease progression. Deriving effective early treatments for these conditions, and better understanding their etiology, likely requires a better understanding of these evolving vascular changes.

However, studying the role of vascular factors in age-related neurodegenerative diseases has been challenging because of difficulties in longitudinally tracking their development. To date, multi-endpoint terminal approaches have been used for determining such vascular alterations at different ages[5–7]. While effective in some respects, these approaches are analytically less efficient, vulnerable to subject-specific random effects, and require an increasing number of animals when involving more measurement time points (Supplementary Fig. 1). Fluorescence multi-photon microscopy has revolutionized the in vivo visualization of neocortical vascular structure and function (flow)[8], but has not met the need for long term, longitudinally repeated

[1]Warren Alpert Medical School, Brown University, Providence, RI 02912, USA. [2]School of Engineering, Brown University, Providence, RI 02912, USA. [3]College of Medicine, Drexel University, Philadelphia, PA 19104, USA. [4]Department of Neuroscience, Brown University, Providence, RI 02912, USA. [5]Department of Electrical Engineering, École Polytechnique de Montréal, Montréal, QC H3T 1J4, Canada. [6]Department of Biostatistics, Brown University School of Public Health, Providence, RI 02912, USA. [7]Department of Molecular Biology, Cell Biology and Biochemistry, Brown University, Providence, RI 02912, USA. [8]Carney Institute for Brain Science, Brown University, Providence, RI 02912, USA. [9]Department of Biomedical Engineering, Boston University, Boston, MA 02215, USA. [10]These authors contributed equally: Konrad W. Walek, Sabina Stefan. ✉e-mail: jonghwan_lee@brown.edu

imaging. Bleaching and phototoxicity[9,10] and the instability of genetically-encoded indicators limit the use of fluorescent methods for stable repeated measurements over several months[11–14].

Optical coherence tomography (OCT) is a label-free technology that provides several unique imaging capabilities without the aid of fluorescence[15–17]. Because of its sensitivity to moving red blood cells (RBCs), OCT can rapidly produce microangiograms[18], quantitatively measure blood flow of individual vessels[19], and even detect passage of individual RBCs in capillary vessels[20], all in a label-free manner. These capabilities suggest OCT can provide a unique solution to the problem of longitudinally studying aging. However, it is unclear for how long OCT methods can repeatedly and robustly image the same cortical microvasculature, and the impact of aging on this imaging. In addition, it is also important to consider the impact of repeated assessments on the same animal, particularly the potential effect of the chronic cranial window on aging animals. More importantly, studies using fluorescence microscopy or OCT have, to date, only measured a few vascular properties at a time[8,11,12] making it difficult to investigate how degeneration in individual microvascular structural and functional properties distinctly correlate with and/or contribute to etiology of disease. A further constraint on using any method, including OCT, for such imaging is the intensive data integration and management challenges created by longitudinal acquisition. Ideally, such image sets should be processed in an automated, unbiased manner; particularly given their size, it poses another technical challenge to be addressed. As one example, the longitudinal experiment presented in this paper produced more than 50 terabytes of raw data, whose analysis would have been unrealistic without automation.

To meet the methodological need and address the related challenges, we present a set of integrated methods for tracking and analyzing diverse properties of the morphology, topology, and function of cortical microvasculature. The approach invented for this purpose and detailed here provides simultaneous measurement of 25 vascular properties across all scales, ranging from large pial vessels to penetrating arterioles and venules to capillaries. For this comprehensive set of properties, a variety of techniques were adopted, developed, and integrated, including OCT microangiography[18], Doppler OCT[19,21], red blood cell (RBC) passage measurements[20], and several deep-learning toolkits[22] (Supplementary Text 1). This integrated set of imaging techniques and image processing algorithms has enabled near-lifespan longitudinal tracking of the vascular features in the aging brain. The present paper demonstrates this methodological capability in 3xTg model mice of Alzheimer's disease (AD) with wild-type (WT) controls, by quantifying a process of cerebral microvascular degeneration over the course of seven months. When applied to broad age-related neurodegenerative diseases, the presented method framework is anticipated to facilitate findings on the roles of microvascular factors in etiology and pathophysiology of disease.

## Results

### Structural and functional properties of pial and penetrating vessels

We repeated microscopic brain imaging sessions every four weeks through a chronic cranial window under anesthesia (7 AD and 6 WT mice, see Methods for details). Figure 1 displays an example set of OCT angiograms obtained from the same animal cortex across seven months. It demonstrates that the label-free, in vivo microscopy technique is stable enough for longitudinally imaging the cortical microvasculature for a long period of such duration. To quantitatively track diameter changes over the same set of pial vessels from the acquired images, we registered the images for each animal (Fig. 1a), selected a set of the same vessels across ages (Fig. 1b), and then measured the diameter of each vessel by fitting its cross-sectional profile to a Gaussian function (Fig. 1c). This allowed us to longitudinally track the same set of 107 pial vessels of 13 mice over seven months

(gray lines in Fig. 1d). These measures were analyzed by linear mixed-effects (LME) fitting to either a linear or nonlinear (sigmoidal) model, where each value was normalized by the group baseline to focus on relative changes with aging and how the changes differed between the AD and WT groups ("Methods"). LME analysis revealed that in AD, average pial vessel diameter decreased with age at a rate of 1.3% per month (95% confidence interval [CI], −2.4 to −0.2), but this rate was not significantly different from WT mice (−1.1% per month; 95% CI, −2.2 to 0.0; $p = 0.78$ between AD and WT; Fig. 1e).

Every imaging session included acquisition of Doppler OCT images as well. We used these images to track changes in the diameter and blood flow of individual penetrating vessels (Fig. 2). The diameter of each vessel was measured by fitting its *en-face* cross-section to a 2D Gaussian function, while the blood flow was measured by the area-integral method[19] after noise reduction[21]. This enabled us to longitudinally track the same set of 151 penetrating vessels of 13 mice for 7 months (gray lines in Fig. 2c, f).

Arteriolar diameter showed sigmoidal decreases in both AD and WT but faster in AD, with the rates of change defined from the sigmoid fits (Methods) being −13% per month in AD (95% CI, −18 to −9) and −7% per month in WT (95% CI, −11 to −3; $p = 0.008$; Fig. 2c). In turn, their fractional changes became significantly different between AD and WT at 18 weeks of age (WOA) ($p < 0.05$, Fig. 2d), which is termed as the age of significance (AOS) hereafter. Second, the venular diameter decreased in AD with the rate of change of −13% per month (95% CI, −18 to −8), significantly different from WT (1.3% per month; 95% CI, 0.9–1.7; $p < 0.001$), leading to the AOS of 12 WOA (Fig. 2e). Third, the arteriolar flow decreased in both AD and WT but faster in AD (−13% versus −10% per month, $p < 0.001$; AOS, 21 WOA; Fig. 2g). Fourth, the venular flow decreased in AD only (−16% per month; 95% CI, −25 to −8; AOS, 18 WOA; Fig. 2f, h). Finally, the penetrating vessel density in vessel number per unit area decreased in AD (−6% per month; 95% CI, −11 to −1), but the slopes were not significantly different between AD and WT ($p = 0.11$). It is interesting to see the AOS be earlier for the structural degenerations than their flow counterparts (see Discussion for detailed interpretation).

### Structural and functional properties of capillary vessel networks

To longitudinally track alterations in smaller vessels like capillaries, every imaging session also included acquisition of another OCT angiography dataset with a higher spatial resolution (3 μm) and an optical focus on the cortical capillary bed ("Methods"). To extract as much information as possible from these microangiography images, we first converted grayscale microangiograms into a graph of the capillary vessel network, using our deep learning-based toolbox for enhancement, segmentation and vectorization of vessels[22] (Fig. 3a). From these network graphs and the original grayscale images, our methods enabled measurements of numerous angio-architectural properties, including morphological properties like the length, diameter, and tortuosity of each vessel; and topological properties such as the branching order, betweenness, closeness, and shortest cycle of each vessel. Betweenness, closeness, and the shortest cycle are used in graph theory for elucidating how nodes are connected within a network (see Supplementary Fig. 4 for conceptual illustration and their implication when applied to a vascular network). While these properties were measured for each capillary segment (108,964 segments in total across 13 animals and seven ages of measurement), we calculated the mean and heterogeneity of every property per animal per age and then tracked them as a function of age, where the heterogeneity was quantified by the coefficient of variation (COV; the standard deviation divided by the mean). Other related measures include capillary number density, capillary length density, and fractal dimension (see Table 1 for a full list).

Prior to analyzing how the properties vary with age, we compared our measures to those reported in the literature when available. The

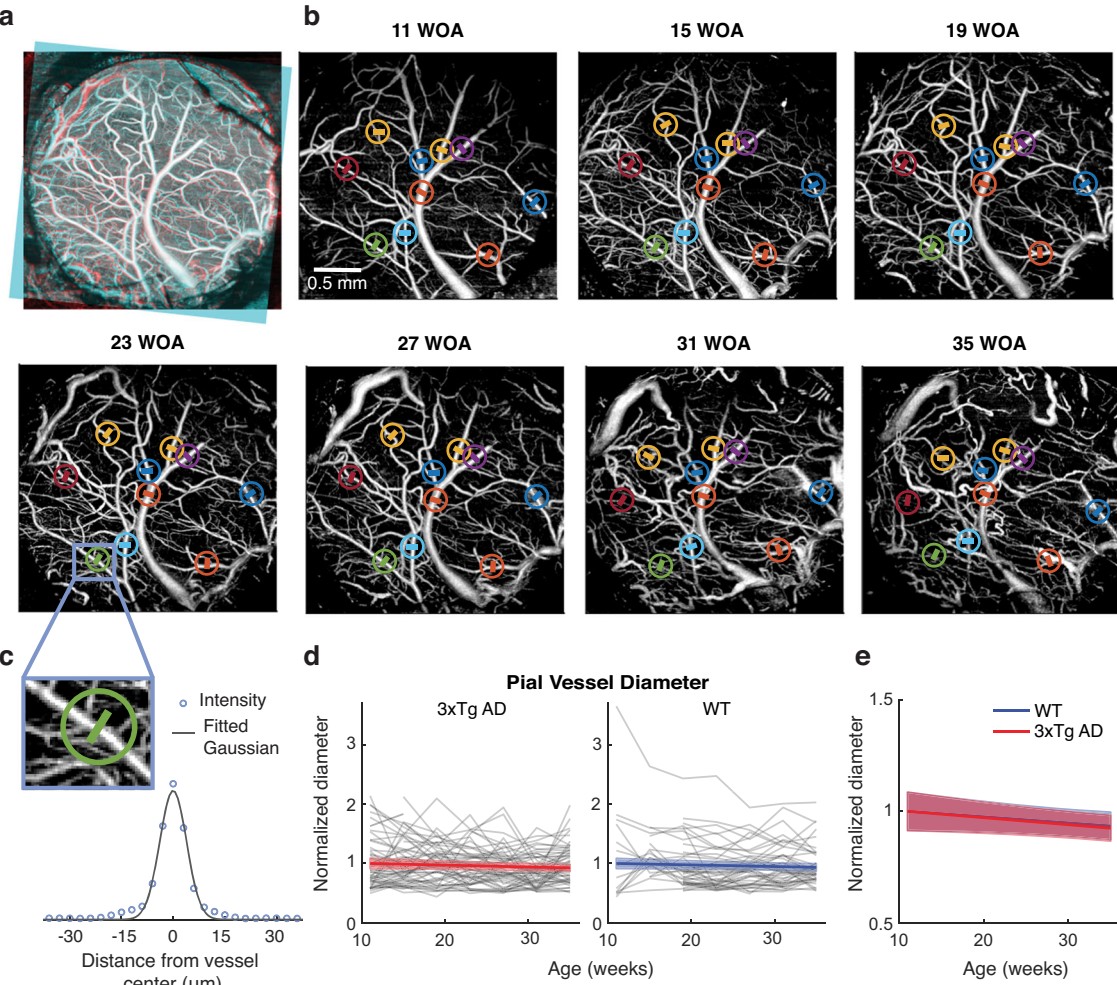

**Fig. 1 | Longitudinal imaging and diameter measurement of pial vessels. a** An example of registration of an angiogram (cyan) to a reference angiogram (red) with overlapping regions shown in white. To achieve this, we selected one angiogram as a reference out of the seven angiograms and used our code to shift and rotate the other six to align their imaging area and angle with the reference. This registration enabled us to visually identify and mark the same vessels across all seven angiograms. **b** An example of selecting the same vessels across time points. The color lines in circles indicate the selected vessels. To ensure selecting the same vessel across time points, we considered its relative position within the vascular branches, as visually shown in this example. Image registration, as shown in (**a**), facilitated this visual inspection and vessel selection. Each of the color lines is drawn along the automatically detected line orthogonal to the orientation of the selected vessel,

along which the cross-sectional intensity profile was extracted for diameter measurement. WOA, weeks of age. **c** An example of the cross-sectional profile and its fitting to a Gaussian function to measure the diameter as the full width at half maximum. To make the diameter measurement robust against slight fluctuations in vessel thickness along the vessel centerline, ten adjacent cross-sections were extracted around the selected location and then averaged along the vessel centerline, prior to the Gaussian fitting. **d** Time courses of the normalized vessel diameters (gray; 60 vessels from 7 AD mice, 47 vessels from 6 WT mice, 7 time points) and their LME fits (color). **e** The LME fits shown together. Angiograms in (**a**) and (**b**) are shown as maximum intensity projection (MIP). The color lines and shades in (**d**) and (**e**) indicate the mean and 95% confidence interval of the LME fits, respectively.

distributions of capillary lengths, diameters, branching orders, and tortuosity obtained from our WT group (Fig. 3b) looked similar in shape and mean value to those previously reported from WT mice[23–26]. We also observed agreement with the literature in the capillary length density and the shortest cycle (Supplementary Tables 1 and 2).

LME analysis revealed several differences emerging between AD and WT. Mean capillary length steadily decreased in AD mice only (−1.5% per month; 95% CI, −2.6 to −0.3; AOS, 25 WOA; Fig. 3c, d). Mean capillary tortuosity also significantly increased in AD only (2.2% per month; 95% CI, 1.3–3.1), but its fractional changes did not become significantly different between AD and WT until the end of measurement (35 WOA). Regarding network topology, the shortest cycle increased in AD only (1.9% per month; 95% CI, 0.3–3.4; AOS, 20 WOA; Fig. 3e). The betweenness also increased in AD only (3.5% per month; 95% CI, 1.3–5.8; AOS, 21 WOA; Fig. 3g). In contrast, the closeness did not change significantly with aging in both AD and WT (Supplementary

Fig. 3), but heterogeneity (COV) decreased in AD (−3.0% per month; 95% CI, −7.9 to −2.0) whereas it increased in WT (3.6% per month; 95% CI, 0.9–6.4), with an AOS of 18 WOA (Fig. 3i).

To assess the pattern of blood flow through a large microvascular network involving hundreds to thousands capillary vessels, we previously presented a method for high-throughput measurements of red blood cell (RBC) flux based on OCT detection of RBCs passing through a voxel[20]. However, this method tends to underestimate high flux values[27] and requires user-selected thresholds. To overcome these weaknesses and improve accuracy, we developed a deep learning-based method by using two sets of RBC-passage data that were simultaneously obtained with OCT and two-photon microscopy ("Methods"). This new method significantly outperformed the previous method (Fig. 4a, b, Supplementary Text 2).

To examine long-term changes in capillary network blood flow, every imaging session included OCT acquisition of RBC-passage data,

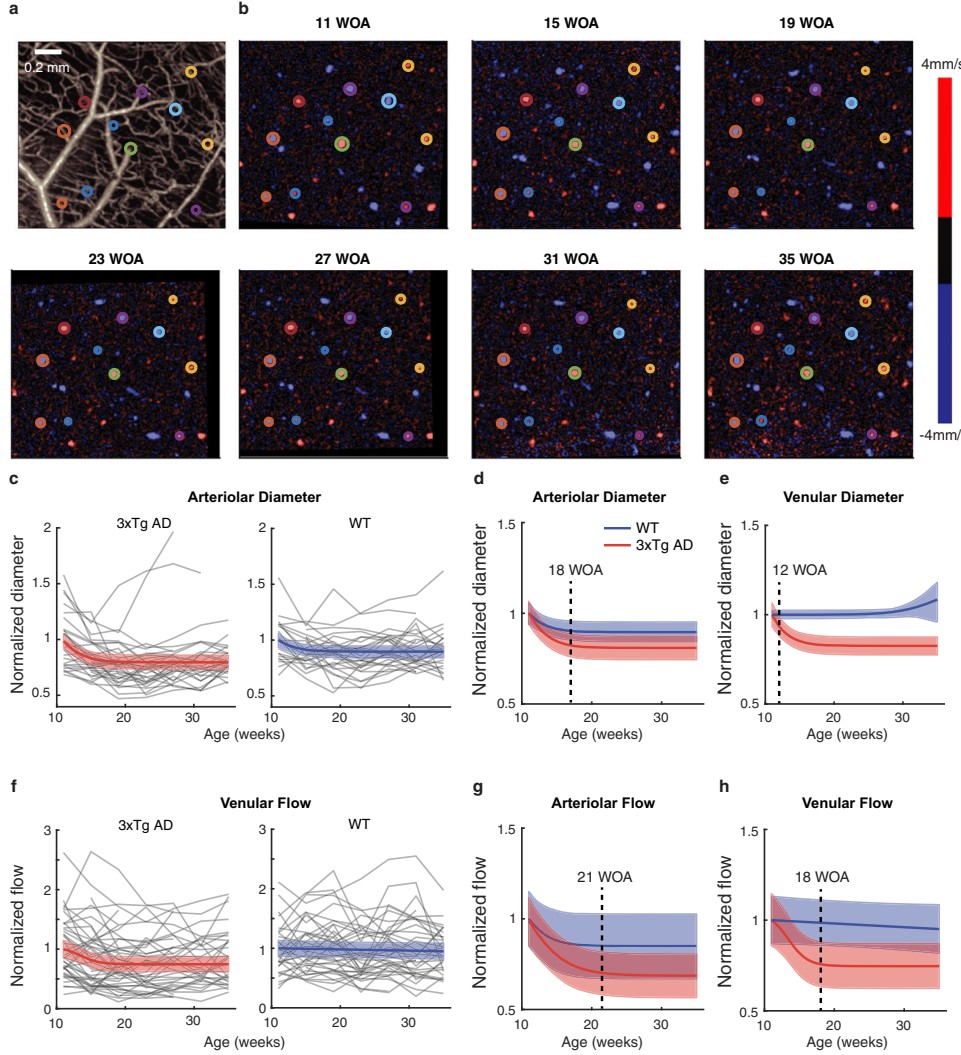

**Fig. 2 | Longitudinal monitoring of blood flow and diameter in penetrating arterioles and venules. a** An example of angiogram MIP with overlaid circles denoting the positions of the penetrating vessels tracked in (**b**). **b** An example set of *en-face* slices of the Doppler velocity maps acquired from the same animal for seven months. Eleven of the same vessels were selected in this animal (circled). WOA, weeks of age. **c** Time courses of arteriolar diameter of individual vessels (gray; 36 vessels from 7 AD mice, 29 vessels from 6 WT mice, 7 time points) and their LME fits

(color). **d** The LME fits of arteriolar diameter shown together. **e** LME fits of venular diameter. **f** Time courses of venular flow of individual vessels (gray; 47 vessels from 7 AD mice, 39 vessels from 6 WT mice, 7 time points) and their LME fits (color). **g** LME fits of arteriolar flow. **h** The LME fits of venular flow. The color lines and shades in (**c**–**h**) indicate the mean and 95% CI of the LME fits, respectively. and dotted lines indicate AOS. See Supplementary Fig. 2 for individual vessel traces of venular diameter and arteriolar flow.

and the deep learning method was used to produce a 3D map of capillary RBC flux values (Fig. 4c). As a result, WT mice exhibited a gradual increase with aging (2.1% per month; 95% CI, 0.9–3.2; Fig. 4e). This increasing trend agrees with previous findings from WT rats[28] although the previous study measured the flux values only at two ages in a non-longitudinal manner. Interestingly, AD mice also showed an increase in flux with aging but at a significantly higher rate (6.0% per month; 95% CI, 3.0–9.0; $p < 0.001$ against WT), making the AOS to be 22 WOA (Fig. 4f). The capillary flux heterogeneity (COV) decreased in both AD and WT but faster in AD (−6.7% versus −3.0% per month; $p < 0.001$; AOS, 23 WOA; Fig. 4g).

### Temporal relationships between microvascular degenerations and cognitive impairment

To simultaneously obtain the time course of cognitive impairment, the animals also underwent novel object location (NOL) tests every month (Methods; the NOL test assesses spatial cognition and memory[29]). While WT mice showed no significant changes in the discrimination index of NOL, AD mice showed cognitive decline with the discrimination index

becoming significantly lower than WT at 27 WOA (Fig. 5a, b), consistent with previous findings from the identical 3xTg AD model[30].

As listed in Table 1, 10 out of 25 vascular properties showed significant differences in fractional changes between AD and WT during the ages of measurement (11–35 WOA). To illustrate the temporal relationships of these vascular degenerations to the cognitive impairment, we plotted a chronological graph as shown in Fig. 5c. This graph revealed three interesting relationships. First, all the observed vascular degenerations preceded the cognitive decline, up to 15 WOA early (about a seventh of the typical lifespan). Second, degenerations in arterioles and venules appeared earlier than those in capillary vessels. Third, structural changes generally preceded corresponding flow changes (arteriolar/venular diameter versus flow, and capillary betweenness/shortest cycle versus RBC flux; see Discussion for interpretation).

To reveal the correlation between the simultaneously observed vascular alterations and cognitive impairment, we calculated the correlation coefficient with age lags of 0, 4, 8 and 12 weeks in the AD group (Methods). Our analysis found that 19 vascular properties were

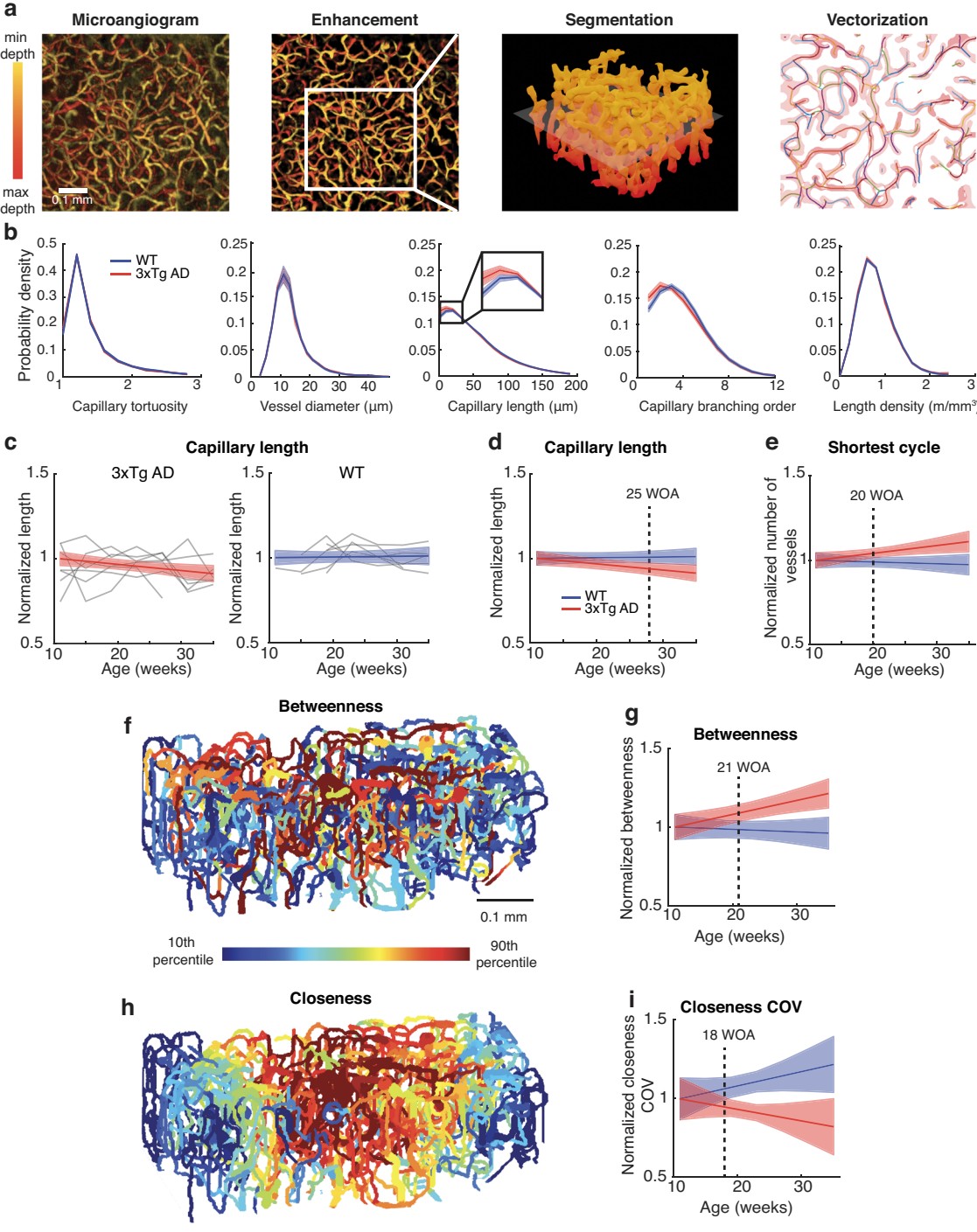

**Fig. 3 | Longitudinal monitoring of microvascular morphological and topological properties. a** An example of the deep learning-based image processing pipeline results, from the input of microangiogram to its enhancement, vessel segmentation (all shown in MIP), and the vectorized capillary vessels. **b** Probability density distributions of capillary tortuosity, vessel diameter, capillary length, branching order, and capillary length density. The distributions were gathered from all animals and all measured ages, for each group. **c** Time courses of the normalized mean capillary length (gray; 7 AD mice and 6 WT mice, 7 time points) and their LME fits (color). **d** The LME fits of the mean capillary length shown together. **e** LME fits of the mean shortest cycle. **f** An example of a 3D betweenness map from an animal at one age, with each vessel colored according to its betweenness value. **g** LME fits of betweenness. **h** An example of a 3D closeness map of the same capillary network as (**f**). **i** LME fits of closeness COV. The color lines and shades indicate the mean and 95% CI, respectively, and dotted lines indicate AOS. See Supplementary Fig. 3 for all properties whose fractional changes did not become significantly different between AD and WT until the end of measurement (35 WOA). WOA weeks of age.

significantly correlated with the NOL discrimination index (Fig. 5d). Some correlations were expected, such as the positive correlations between the decrease in NOL test score and the decreases in arteriolar and venular diameter and flow. Other correlations were unexpected; for example, the betweenness was strongly negatively correlated with

the NOL score. This betweenness result was interesting when considering that it exhibited the youngest AOS among capillary vessel properties (Fig. 5c) and formed the greatest number of inter-correlations with other vascular alterations (Supplementary Fig. 8, see "Discussion" for interpretation).

**Table 1 | Summary of statistical results of 25 vascular properties**

| Property | Rate of change with age (RCA) in AD (%/month) | RCA in WT (%/month) | P value of difference in RCA between AD and WT | The age of significance (weeks) |
|---|---|---|---|---|
| Pial vessel diameter | −1.3 (−2.4 to −0.2) | −1.1 (−2.2 to 0.0) | 0.78 | |
| **Arteriolar diameter** | **−13.2 (−17.5 to −8.7)** | **−7.2 (−11.4 to −2.8)** | **0.008** | **18** |
| **Venular diameter** | **−13.1 (−18.0 to −7.8)** | **1.3 (0.9–1.7)** | **p = 2.4e-8** | **12** |
| Penetrating vessel density | −5.7 (−10.8 to −0.6) | −1.3 (−6.7 to 4.7) | p = 0.11 | |
| **Arteriolar flow** | **−12.9 (−18.7 to −7.3)** | **−9.6 (−19.4 to −0.6)** | **p = 1.4e-6** | **21** |
| **Venular flow** | **−16.3 (−24.9 to −7.9)** | **−0.8 (−2.9 to 1.3)** | **0.009** | **18** |
| **Capillary length** | **−1.5 (−2.6 to −0.3)** | **0.2 (−1.0 to 1.3)** | **0.005** | **25** |
| Capillary length COV | −0.2 (−1.0 to 0.7) | 0.1 (−0.9 to 1.1) | 0.56 | |
| Capillary diameter | 0.2 (−0.4 to 0.9) | 0.0 (−0.8 to 0.9) | 0.63 | |
| Capillary diameter COV | 0.3 (−0.7 to 1.3) | 0.5 (−0.3 to 1.3) | 0.60 | |
| Capillary tortuosity | 2.2 (1.3–3.1) | 0.7 (−0.5 to 1.9) | 0.015 | >35 |
| Capillary tortuosity COV | 1.0 (−0.1 to 2.1) | 3.4 (−3.5 to 10.4) | 0.60 | |
| Branching order | 0.9 (−0.6 to 2.4) | 1.4 (0.3–2.5) | 0.22 | |
| Branching order COV | 1.3 (0.4–2.3) | 1.0 (0.2–1.9) | 0.49 | |
| **Betweenness** | **3.9 (1.3–5.8)** | **−0.7 (−2.8 to 1.5)** | **p = 0.0003** | **21** |
| Betweenness COV | −1.7 (−3.2 to −0.2) | −0.9 (2.2–0.4) | 0.21 | |
| Closeness | −1.6 (−3.8 to 0.6) | −1.1 (−3.2 1.0) | 0.67 | |
| **Closeness COV** | **−3.0 (−7.9 to −2.0)** | **3.6 (0.9–6.4)** | **p = 7.8e-6** | **18** |
| **Shortest cycle** | **1.9 (0.3–3.4)** | **−0.4 (−1.7 to 0.8)** | **p = 0.0006** | **20** |
| Shortest cycle COV | −0.7 (−2.2 to 0.7) | −1.3 (2.6–0.0) | 0.37 | |
| Capillary number density | 0 (−0.9 to 1.0) | 0.0 (0.0–0.0) | 0.23 | |
| Capillary length density | 0.6 (−0.6 to 1.9) | 0.8 (−0.7 to 2.4) | 0.79 | |
| Fractal dimension | 0.1 (−0.4 to 0.3) | −0.1 (−0.5 to 0.2) | 0.66 | |
| **RBC flux** | **6.0 (3.0–9.0)** | **2.1 (0.9–3.2)** | **p = 3.7e-9** | **22** |
| **RBC flux COV** | **−6.7 (−11.4 to −2.1)** | **−3.0 (−4.7 to −1.4)** | **p = 2.0e-5** | **23** |

The two numbers in the parenthesis present 95% CI. See "Methods" for how the rate of changes with age (RCA) was defined when the fractional change of a property was better fit with the sigmoid function.

Shown in bold are those that showed significant differences between AD and WT before the latest age of measurement (35 WOA, significance tested with α = 0.05 after Benjamini–Hochberg correction; p values were obtained using a Wald test).

## Discussion

We have demonstrated that the integrated set of label-free in vivo imaging and image-processing methods produces an unprecedented set of data regarding age-related alterations in the morphology, topology, and function of the cortical vasculature across all scales. Such information-rich, multi-feature, longitudinal datasets allow to simultaneously investigate multiple aspects of slowly developing cerebral microvascular degeneration while affording high statistical power (Supplementary Fig. 1). Among other results presented here, the chronological and correlation graphs between cognitive impairment and vascular alterations (Fig. 5c, d) are a representative example of what type of information can be obtained by the presented methods. Such information may provide insight into how cognitive impairment and different types of vascular alterations develop along with or independently of each other. Moreover, the presented methods align with the "3R's" principles of animal research, particularly reduction. For example, while a vascular corrosion casting study demonstrated age-related changes in capillary morphology, it required more animals due to the terminal nature of the method[5]. Similarly, another casting study identified that changes in capillary morphology preceded cognitive decline, but it was limited to investigating only one age point out of a similar number of animals[31]. In contrast, the in vivo nature of the OCT method used in this study provides a clear advantage in reducing the number of animals required, while simultaneously providing detailed information about the dynamics of cerebrovascular changes.

In the presented demonstration, we stopped the longitudinal imaging when the AD mice exhibited obvious differences from WT in the cognitive function test for two consecutive sessions (i.e., for eight weeks). The tracking period was approximately a third of the typical mouse lifespan: The potential tracking period could be even longer as the image quality did not significantly degrade over time in many animals (Fig. 1b for example). One limitation in the presented demonstration is that we conducted all imaging experiments under anesthesia, although anesthesia unlikely affects our major outcome, i.e., difference in the rate of change with age (RCA) between AD and WT (see Supplementary Text 9 for related data and discussion). The presented methods are readily applicable to awake imaging as being shown in our follow-up studies. In the context of longitudinal assessments of awake animals, our methods can offer detailed, microvasculature-level assessments under a head-fixed condition. In comparison, diffuse correlation spectroscopy has shown the ability to provide assessments under a freely behaving condition but at a less detailed, macroscopic level[32]. Combining these two methods could yield a more comprehensive understanding of the changes and mechanisms underlying cerebrovascular dysregulation in Alzheimer's disease and other progressive vascular diseases. Supplementary Text 3 provides further discussion of other challenges and opportunities related to the presented methods. Supplementary Text 10 discusses the sensitivity of our approach to detecting changes in vascular diameter and blood flow. Supplementary Fig. 7 summarizes how the various types of OCT data went through the described image-processing pipeline, and all processing software codes are publicly opened (see "Code availability").

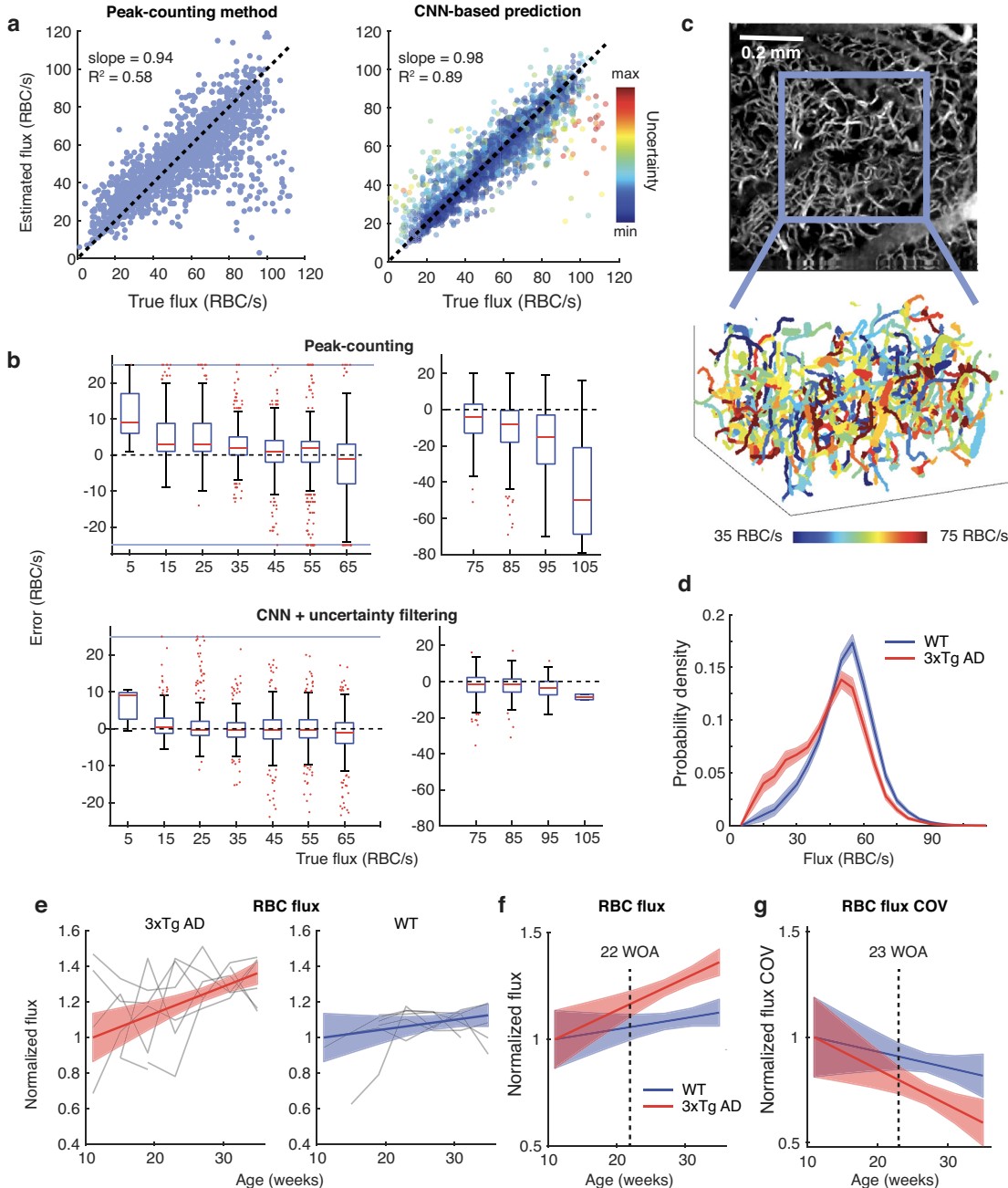

**Fig. 4 | Longitudinal tracking of capillary network RBC flux. a** Compared to the traditional peak-counting method, the convolutional neural network (CNN)-based method produced the slope closer to 1 and higher R² (coefficient of determination). The uncertainty of individual estimations of the CNN-based method are color-coded in the right figure, while the traditional peak-counting method does not provide such uncertainty of estimation (Supplementary Text 2). **b** Distribution of errors of the two methods across different flux ranges (1,888 observations from 119 capillaries across 3 animals). When the CNN-based method filtered out predictions with the lowest 20% confidence, the error became both smaller (narrower distribution) and less biased (the mean closer to 0 RBC/s) even for higher flux values. Each box chart displays the median (red line), the lower and upper quartiles (box),

outliers (red dots, computed using the 1.5× interquartile range), and the minimum and maximum values that are not outliers (whiskers). Source data are provided as a Source Data file. **c** An example of microangiogram constructed from RBC-passage data (top) and the vascular graph vectorized from the microangiogram (bottom). **d** Distribution of flux values gathered from all animals and all ages. **e** RBC flux changes as tracked through the same animals (gray; 7 AD mice and 6 WT mice, 7 time points) and their LME fits. **f** The LME fits of RBC flux shown together. **g** LME fits of the RBC flux COV. The color lines and shades in (**d**–**g**) indicate the mean and 95% CI, respectively, and dotted lines indicate AOS. RBC red blood cell, COV coefficient of variation.

In our results with the 3xTg model (see Supplementary Text 4 for use of the specific model within the scope of this study), the earliest cerebral microvascular degeneration (CMD) was detected in the diameter of penetrating vessels (thinner in AD mice), followed by degeneration in their blood flow (lower in AD). It is interesting to see the structural degenerations precede the flow counterparts (Fig. 2). We speculate that once penetrating vessels start becoming pathological in

structure (i.e., becoming statistically different from age-matched WT), a compensating adaptation occurs to autoregulate cerebral blood flow. However, this adaptation may not be sustained indefinitely, and eventually blood flow in penetrating vessels decreases below the normal physiological range (95% CI of age-matched WT; see Supplementary Text 5 for further discussion). Although the primary focus of our study is technical demonstration rather than biological discovery

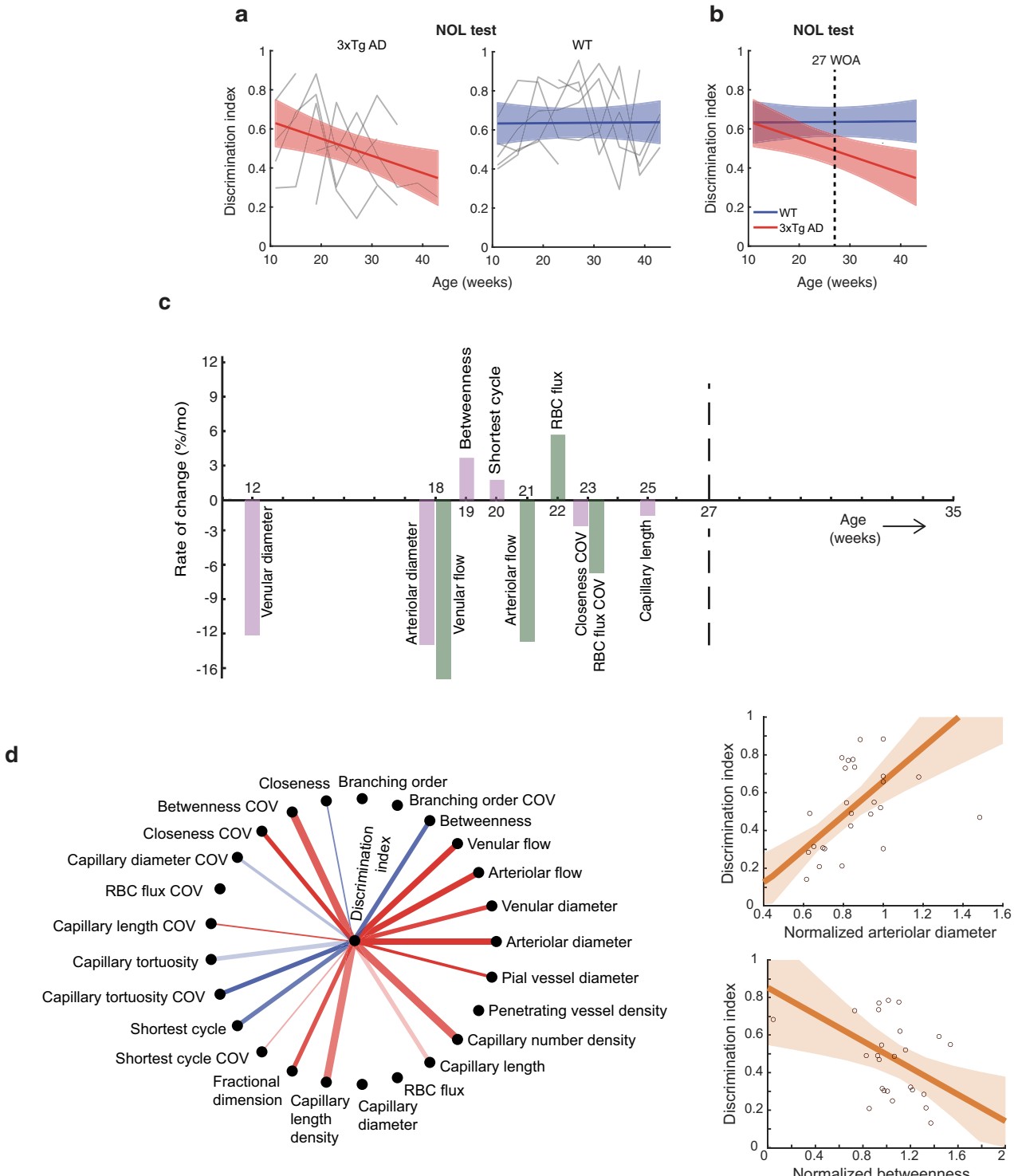

**Fig. 5 | Cognitive testing result, and integrative analysis of the measured vascular properties.** **a** NOL test scores tracked by individual animals (gray lines; 7 AD mice and 6 WT mice, 9 time points) and their LME fits (color line, mean; color shade, 95% CI). **b** The LME fits shown together (line, mean; shade, 95% CI). **c** Chronological graph of all vascular properties whose fractional changes became significantly different between AD and WT mice during the ages of measurement. The graph displays their rates of change in AD at their AOS (structural properties in purple, and flow properties in green). **d** Significant correlations between the NOL discrimination index and the vascular alterations ($p < 0.05$, Wald test, no multiple

comparison correction was used; see "Methods" for details). Pairs with positive and negative correlations are shown in red and blue respectively. More opaque lines indicate higher maximum correlation, and thinner lines means that the maximum correlation appeared at longer age lags. A correlation with an age lag means that one alteration was correlated to the other one with a certain time delay. Two examples of significant correlations are presented on the right, where each circle represents a measurement from a single animal at a single age point, and the orange line and shade depict the fitted line (mean) and its 95% CI. Source data are provided as a Source Data file.

(Supplementary Text 4), the observed cortical hypoperfusion in AD is consistent with findings from perfusion magnetic resonance imaging and transcranial Doppler ultrasound studies, which have reported hypoperfusion in various cortical regions in early human AD (see refs. 33,34 for reviews) as well as findings from AD model mice[35,36].

On a smaller scale, the presented results revealed degenerations in both morphology (the mean capillary length) and topology (the shortest cycle, betweenness, and closeness COV) of capillary vessel networks. The mean capillary length result (becoming shorter with aging in AD only, Fig. 3d) is consistent with the previous finding that capillary length is shorter in AD than WT[23], although the previous study did not longitudinally track the length and used a different AD mouse model (see Supplementary Text 7 for further discussion). The capillary tortuosity did not show significant difference in its fractional changes between AD and WT until the end of measurement (35 WOA), but its rate of change with age was positive in AD only. While vessel diameter and tortuosity can be correlated (e.g., with thicker vessels being less tortuous), it is unlikely that the observed significance in the tortuosity change rate of AD is an influence of diameter changes because the diameter change rate did not show significant difference between AD and WT (Table 1). In contrast, the topological properties have been little studied in the context of aging. Interestingly, the topological changes became pathological earlier than the morphological change (Fig. 3). Also, both the shortest cycle and the mean betweenness increased with aging in AD, likely indicating a higher degree of susceptibility to vascular insult than WT (Supplementary Text 6). This higher susceptibility may be important in AD when considering the increased rate of capillary stalling in AD models[37]. The presented results also showed degenerations in capillary blood flow (see Supplementary Text 7 for interpretation and potential relation to hypertension).

Finally, the chronological graph showed that most CMDs became apparent between 12 and 25 WOA and preceded the cognitive decline observed from the same animals (Fig. 5c). Some of these CMDs slightly precede even extracellular Aβ deposits in the 3xTg model (see Supplementary Text 11 for details). The correlation graph (Fig. 5d) showed that 19 vascular properties were significantly correlated with the cognitive NOL test score. While some of these confirmed expected correlations, such as those of arteriolar and venular diameter and flow, others revealed new findings. Of particular interest is the betweenness property, which displayed a strong negative correlation with the NOL test score, meaning that higher betweenness was associated with later cognitive decline. It was the earliest capillary vessel property to show significant differences between AD and WT mice (Fig. 5c) and was the most inter-correlated with other vascular alterations in both AD and WT, but with different inter-correlation patterns (Supplementary Fig. 8). These interesting findings suggest that the betweenness of the capillary network is an important component of CMD in neurodegenerative diseases. This spotlight on the relatively novel topological property, now longitudinally measurable in vivo using the presented methods, encourages further studies on how other vascular and non-vascular factors interact with capillary network betweenness and how this interaction contributes to the pathology of vascular and other related systems in aging research.

Aβ clearance plays an important role in AD[38]. Various overlapping and interacting clearance mechanisms are being studied, including the glymphatic pathway. One of these clearance systems, called perivascular system, involves a number of components such as the cerebrospinal fluid, interstitial fluid, periarterial space, water channel aquaporin-4 in astrocytes[39], extracellular space affected by astrocytic endfeet structure[40], and fiber myelination[41,42]. Since many of these components are structurally and functionally interrelated with blood vessels, the methods presented in this paper would be useful in studying how the clearance system components interact with the related vascular properties, how the interaction varies with age, and

how the age-dependent interaction differs between AD and normal aging.

Future findings enabled by the presented methods in diverse AD models, when combined with follow-up studies at cellular/molecular level, may determine whether CMD is etiological in AD pathogenesis, one of the long-lasting questions in AD research[43]. Such findings can suggest updating the AT(N) framework to AT(VN) for the Alzheimer's disease continuum[44], and more importantly, will lead to the development of therapeutic targets as well as biomarkers for preemptive early diagnosis. When widely applied beyond AD, the presented methods are expected to provide new insight into progressive development of vascular etiology and pathophysiology, or a means to monitor long-term vascular responses to therapeutic intervention, in a range of age-related neurodegenerative diseases.

## Methods
### Animal preparation
Male wild-type (WT, C57BL/6J, Jackson Lab) mice ($n = 10$) and male 3xTg AD mice (B6;129-Tg(APPSwe,tauP301L)1Lfa Psen1TM1Mpm/ Mmjax, Jackson Lab, $n = 10$) underwent craniotomy at 10 weeks of age (WOA) for window placement[45] and in vivo OCT imaging under inhaled anesthesia. Of these mice, one mouse died during the experiment, three mice were euthanized due to damage on their headposts, and three mice were excluded due to degrading image quality (see Supplementary Text 8 for potential impact of long-term craniotomy and imaging on animal physiology). The mice were housed in a controlled environment with a 12-h light/dark cycle, with lights on at 7:00 a.m. and off at 7:00 p.m. The ambient temperature was maintained at a constant $22 \pm 2\,°C$, and the relative humidity was kept at $50 \pm 10\%$. Mice were housed in autoclaved HEPA-filtered ventilated cages made of standard polycarbonate with sterilized hardwood (Beta Chip) bedding and provided with environmental enrichment in the form of nesting material and a plastic tunnel. Autoclaved food and water were provided ad libitum, and cages were changed weekly to ensure a clean and hygienic living environment. All experimental procedures involving animals were reviewed and approved by the Institutional Animal Care and Use Committee (IACUC) of Brown University and Rhode Island Hospital. Experiments were conducted according to the guidelines and policies of the office of laboratory animal welfare and public health service, National Institutes of Health.

**Anesthesia and perioperative monitoring.** At 10 WOA, mice underwent induction of anesthesia with 3% isoflurane in oxygen and were then maintained at 2% isoflurane until recovery in 100% oxygen. For both the initial surgery and subsequent imaging sessions, the duration of anesthesia from induction to recovery was approximately 2 h. Oxygen saturation, pulse rate, and temperature were continuously monitored with a pulse oximeter and rectal probe for the entire surgical procedure and imaging sessions. The body temperature was maintained at 37 °C and the pulse remained within the normal range of 250–350 bpm.

**Craniotomy.** After induction of anesthesia, the scalp overlying the parietal bones was shaved and then sterilized using alcohol and iodine scrub. A 1-cm midline incision was made on the scalp, and the skin on both sides of the midline was retracted. The pericranial tissue was stripped using needle-tip forceps. A proprietary metal frame, used to hold the head and prevent motion during microscopy (Supplementary Fig. 11), was affixed to the skull with dental cement (Parkell Inc., Long Island, NY, USA). A 3-mm bone flap overlying the left parietal cortex was thinned with a dental burr until transparent (100-μm thickness). In detail, the location of the cranial window was identically selected in all mice, such that the medial border of the window was 1 mm lateral to the sagittal suture and the posterior border was 1-mm anterior to the lambdoid suture on the right side, allowing for visualization for visual,

somatosensory, and motor cortex. The thinned bone flap was removed with fine-tip forceps while keeping the dura intact and the skull defect filled with normal saline until hemostasis was achieved. Our circular glass cranial window detailed below was then placed into the defect. The window was then fixed in place and the annular skull defect surrounding the window was sealed using dental cement. The cement was given time to fully cure, and the mouse was then recovered in an oxygen chamber. Mice were housed in individual cages post-operatively to protect the integrity of the metal frame.

The base of the window consisted of two 3-mm glass cover slips and the apex consisted of one 5-mm circular glass slip; these were epoxied together with a transparent UV-cured resin. The thickness of the 3 mm-wide base of the window was 0.24-mm (two stacked cover slips of 0.12-mm thickness each), closely approximating the thickness of the mouse parietal bone (~0.2 mm)[46]. The 5-mm cover slip serving as the apex of our window acted as an additional safety measure against compression of the cortex, as it was wider than our 3-mm craniotomy and thus any pressure against the window prior to cementing was applied to the bone surrounding the craniotomy and not the brain tissue itself; it also ensured that the base of the window sat flush just above the cortex without compressing it. Once the window was cemented in place, this minimized any dural scarring or scar tissue ingrowth underneath the window for the duration of our experiment.

The use of an open-skull cranial window has clear strengths and weaknesses against a thinned-skull cranial window. The open-skull method in this study provides higher quality microangiography where individual capillaries are clearly visualized (Fig. 3a) and individual RBC passages through those capillaries are clearly captured in the time courses of the image intensity signal (Supplementary Fig. 5). However, the open-skull method is known to be relatively less robust in a long term; we began with 20 animals, but one animal did not survive until the end of our seven-month longitudinal experiment. In three out of the surviving animals, the imaging quality through the window degraded significantly with time, secondary to scar tissue formation underneath the window such that insufficient number of vessels were visible with imaging—these animals were not included in our analysis. Additionally, three animals were euthanized in the middle of experiment due to headpost damage. We also note that installing a chronic cranial window in mice at younger than 10–12 weeks can be problematic due to incomplete skull growth prior to this age[47].

## OCT imaging
All OCT imaging was performed in vivo. Baseline OCT imaging was obtained seven days following the surgery, and subsequently each animal underwent imaging every four weeks for seven months. After induction of anesthesia and for the duration of imaging, the mouse's head was stabilized against motion by attaching the affixed metal frame to a proprietary platform with a heating pad and inhaled iso-flurane delivery system underneath the OCT system.

All OCT measurements were collected with a commercial SD-OCT system (Thorlabs, Newton, NJ, USA). We used custom LabVIEW software to control the OCT system. The system uses a large-bandwidth near-infrared light source with a center wavelength of 1310 nm, and wavelength bandwidth of 170 nm, which leads to a high axial resolution of 3.5 μm. The system uses a high-speed 2048-pixel line-scan camera to achieve 147,000 A-scans/s. During each imaging session, we acquired angiograms of large pial vessels using a 5× objective lens, with a field of view (FOV) of 1024 × 1024 × 512 (x,y,z) corresponding to an imaging volume of about 3 × 3 × 1.8 mm. Ten angiogram volumes were acquired, motion-corrected[48], and then volume-averaged prior to diameter measurements, in order to minimize the effect of physiological cardiac fluctuations. Microvessels were imaged using a 10X objective lens with a FOV of 1024 × 1024 × 512 (x,y,z) corresponding to an imaging volume of about 1.5 × 1.5 × 1.8 mm (x,y,z). We measured blood flow in penetrating vessels using Doppler OCT by repeating 8

A-scans at each (x,y) position over a FOV of 512 × 512 × 512 (x,y,z) corresponding to a region of about 1.5 × 1.5 × 1.8 mm. RBC flux data consisted of 512 repeated B-scans with a temporal resolution of 1.4 ms, producing a 4D dataset of size 256 × 256 × 512 × 512 (x,y,z,t), corresponding to an imaging volume of 0.8 × 0.8 × 1.8 mm (x,y,z) over a time period of 0.7 s. Each OCT imaging session took about an hour per mouse; 10 min for angiogram, 5 min for Doppler, 10 min for micro-angiogram, and 30 min for RBC data. We held imaging sessions routinely during the daytime.

## Microangiography analysis
We measured the capillary length and diameter from the segmented image and vectorized graph (Fig. 3a as an example) by measuring the length of the centerline of each capillary segment and the thickness of the segment in the segmented image (see ref. 25 for further details), the histograms of which are shown in Fig. 3b. We also measured the capillary tortuosity from the mean curvature of the vessels, calculated by the mean derivative of the tangent along the vessel[49]. Since tortuosity and diameter can be correlated (e.g., with thick vessels being less tortuous), diameter results should be carefully considered when interpreting tortuosity results. To determine branching order, we leveraged the fact that penetrating vessels appear as dark circles (or shadows) in the structural intensity image. This allowed us to label all penetrating vessels and trace connected vessels as they branched out, thereby finding the closest distance between each capillary and a penetrating vessel along the capillary bed.

Closeness and betweenness aim to elucidate how vessels are connected within the network. Closeness is calculated as the reciprocal sum of the number of vessels along the shortest paths between the vessel and every other vessel in the network. Betweenness counts how many times a single vessel is traversed along the shortest path between any other two vessels within the network (Supplementary Fig. 4). However, these metrics depend on the size of the network, so to account for this we divided the mean values by the mean closeness or betweenness of an idealized honeycomb network with the same number of vessels. Supp. Fig. 4d shows the relationship between mean closeness and betweenness and number of vessels in a honeycomb network, which was used to normalize the values obtained from the experimental data. The COV was normalized in a similar manner.

Lastly, we looked at global properties of the network within the imaging volume: we calculated the capillary number density by dividing the number of capillaries by the volume, and the capillary length density by summing the lengths of all capillaries and dividing by the volume. Fractional dimension was determined using the box-counting method[50].

## Capillary RBC flux measurement
**Data preparation.** We developed the CNN-based, RBC flux-measuring method by utilizing two sets of RBC-passage data that were simultaneously obtained with OCT and two-photon microscopy (TPM). We trained a 1D CNN using the OCT time traces as the training data, and the corresponding RBC flux determined from the TPM time traces as ground truth (Supplementary Fig. 5 and Supplementary Text 2). We augmented the training dataset by reversing the time of the time traces, and split this augmented dataset into training, validation and testing as 70%, 20 and 10% respectively.

**CNN architecture and training.** We implemented the state-of-the-art architecture in time-series classification, InceptionTime[51], as shown in Supplementary Fig. 6. We designed our CNN to provide an estimate of uncertainty in its prediction by adapting the loss function as shown in Eq. 1 to maximize the probability of a given prediction $\hat{y}$, assuming this prediction is derived from a normal probability distribution with mean equal to $G$, and standard deviation, $\sigma$. $G$ is the ground truth, and the standard deviation thus serves as an indicator of uncertainty in the

predicted value. We trained our network using the Adam optimizer, with a learning rate of 0.0008 for 50 epochs, and a gradient threshold of 500 which was necessary given our adapted loss function.

$$loss = -\log\left(\frac{1}{\sqrt{2\pi}\sigma}\exp\left(-\frac{(G-\hat{y})^2}{2\sigma^2}\right)\right) \quad (1)$$

**RBC flux estimation.** To track long-term changes in the capillary network blood flow pattern, we acquired 4D data $(x,y,z,t)$ consisting of 512 time points over a period of 0.72 s. We first constructed an angiogram from this data (Fig. 4c) and obtained a vascular network graph from the image as described in Fig. 3a. We extracted the time course from the 4D data for each centerline voxel of the graphed vessels, where RBC passage is known to cause a transient increase in intensity. We then applied the CNN to each time course along the centerline and averaged the predicted flux values along the centerline voxels for each vessel by weighting the predictions according to their associated uncertainties, thus determining the average RBC flux for each vessel (Fig. 4c, bottom). Finally, we determined the mean flux for each dataset (per animal per time point) by averaging the flux values over all vessels, and the COV by dividing the standard deviation by the mean.

### Novel object location test
Every 4 weeks from 11 weeks of age, each mouse was cognitively evaluated using a novel object location test that is widely used to assess spatial memory[52]. Our test used an open-field apparatus consisting of a square arena $(40 \times 40 \times 49 \text{ cm}^3)$. The mouse was habituated to an empty arena with four visually distinct quadrants for 5 min, and then removed. The arena was cleaned with 70% ethanol and dried to eliminate any potential odor cues left beforehand. Then, two identical objects were placed into the northwest (NW) and northeast (NE) quadrants, while the other quadrants remained empty. The mouse explored the arena for 5 min while the video tracked its movement. The mouse was removed, the arena was again cleaned, and both objects were placed again in the NW and southeast (SE) quadrants such that one of the objects was placed in a different location (SE). After 20 min, the mouse was again placed into the arena and allowed to explore the arena for 5 min while its movement was tracked. Differences in movement between the first and second 5-min tracking were used to assess recognition of the novel object's location.

To determine the sign of cognitive decline, we calculated the discrimination index:

$$Z = \frac{T_{SE}}{T_{NW} + T_{SE}} \quad (2)$$

where $T_{SE}$ is the amount of time spent exploring the object in the novel position, and $T_{NW}$ is the amount of time spent exploring the object in the same position. A $z$-score of 0.5 or below was considered indicating that the mice did not remember one object being moved to a novel location, and thus was considered as the sign for cognitive decline[52].

### Statistical analysis
We used the linear mixed-effects (LME) method to fit each set of measured time courses to either a linear model or a nonlinear model following a sigmoidal curve ($y = c + \frac{L}{1+\exp(k(b-\text{age}))}$), and then selected the model with the lowest Akaike Information Criterion (AIC) following the widely used method[53,54]. The AIC also informed our choice of random effects based on the intercept, slope and/or time-group interaction. We normalized each metric for AD and WT groups by the intercept as determined by the fitted model. Outliers were excluded based on Cook's distance: any value with a Cook's distance greater than 3 times the mean was considered an outlier, as practiced in many studies[55,56].

The rate of change was simply defined by the slope for the linear model. For the sigmoid model, we defined the rate of change by measuring 90% of the total changes observed during the ages of measurement (11–35 WOA; from the bottom 5% to the top 5%) and then by dividing the 90% change by the age duration corresponding to the 90% change. The same was done on the 95% confidence intervals of the nonlinear fitting to obtain the confidence intervals for the nonlinear slope. When comparing the rate of change between AD and WT, for linear models, we recorded the $p$ value of the interaction term to determine significant differences between slopes. For nonlinear models, we determined the $p$ value by conducting a $t$ test between the estimated parameters $L$ for each nonlinear model. We selected vascular properties for further analysis which were associated with significant $p$ values at an overall level of $\alpha = 0.05$ after Benjamini-Hochberg correction to control false discoveries. For these metrics, we determined the age at which AD and WT groups differed by assessing the degree of overlap between two confidence intervals (CIs). To account for multiple hypothesis testing using Bonferroni correction, we aimed to find the age at which the groups were significantly different at a rate of $\alpha = 0.05/7$, which in turn leads to the age at which the 93% CIs just touch. In detail, we computed the distance between the lower CI bound of the group with higher values and the upper CI bound of the other group. The age from which this distance becomes larger than zero was selected as the age at which AD and WT groups started to differ.

### Correlation analysis
To study the correlation between cognitive decline and the vascular features studied, we fitted LME models to determine the relationship between the fractional change of each vascular feature and NOL discrimination index. Two such examples are demonstrated on the right of Fig. 5d. We fitted these models while also introducing time-lags of 0, 4, 8 and 12 months. The features which exhibited a significant slope ($p < 0.05$) were considered to be correlated with cognitive function, with the sign of the slope indicating positive or negative correlation.

### Reporting summary
Further information on research design is available in the Nature Portfolio Reporting Summary linked to this article.

## Data availability
The vascular property measurement and novel object location test data generated in this study have been deposited in the Figshare database: https://doi.org/10.6084/m9.figshare.19178885. The raw image data is not available due to its size (larger than 50 terabytes). Source data are provided with this paper.

## Code availability
All codes for image processing and analysis were developed in MATLAB. The MATLAB codes are available for non-commercial use in GitHub (https://github.com/optobrain/adp-1-3xtg-cortex)[57].

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

## Acknowledgements

This work was supported by the National Institute on Aging award R01AG067228.

## Author contributions

K.W. and J.L. conceived of the study. J.L. supervised the study. K.W. acquired main OCT and NOL data. P.J.M. and F.L. acquired the TPM dataset used for training the RBC flux predictor. J.-H.L. acquired supplementary immunofluorescence and OCT data. S.S. and J.L. developed codes for data analysis, and S.S., P.P., J.-H.L., A.H.K., and S.W.P. performed the analysis. S.S. and T.L. performed the statistical analysis. S.S., Y.-W.A.H., D.B., C.M., and J.L. interpreted the results. S.S. and J.L. wrote the manuscript, and all authors revised the manuscript.

## Competing interests

The authors declare no competing interests.
