## [Peer Review File · Nature Communications]

Near-lifespan longitudinal tracking of brain microvascular morphology, topology, and flow in male miceREVIEWER COMMENTS

Reviewer #1 (Remarks to the Author):

In the present work, Walek et al. longitudinally applied and combined multiple optical coherence tomography (OCT) techniques and integrated them with image processing algorithms to examine the microvasculature at different scales in wildtype and 3xTg mice modelling Alzheimer's disease. The elegant approach enabled longitudinal examination between 10 and 38 weeks, with imaging performed every 4 weeks.

Main comments:

-More details about the simulation concerning the comparison between multi-endpoint and longitudinal approaches presented in suppl. Figure 1 should be provided.

-Introduction, lines 48-49: Important to mention is also the impact of repeated assessments on the same animal, especially of the chronic cranial window on aging animals.

-Methods, Animal preparation, lines 365-368 & 403-405: the authors state that installing a chronic cranial window in mice younger than 10-12 weeks can be problematic due to incomplete skull growth prior to this age. Assuming that the mice were 10 weeks old at the beginning of the study, and taking the 7-month-observation period, would lead to an age of approximately 280 days at the end of the study. For male 3xTg mice, the survival rate has been shown to be 80% at this age, while for WT mice, it is almost 100% (Kane AE et al., *Front Aging Neurosci* 2018; 10:172, doi: 10.3389/fnagi.2018.00172). In the present work, 30%-40% of the mice including wildtype mice, died during the longitudinal assessments over the course of 7 months or were excluded due to image degradation. This seems a quite high number, clearly suggesting to me that the window placement might have a long term influence on the physiology of the animals. Histological analyses of the brain of mice at the age of 280 days should be provided, comparing 3xTg and WT mice with and without cranial window, addressing possible side effects of the window.

-It would be quite interesting to compare the results on OCT microangiography obtained here with a more global technique, e.g. perfusion MRI or transcranial Doppler ultrasound. On the other hand, for validation purposes it would have been important to compare the OCT results with histological analysis of microvasculature, e.g. using collagen IV and transglutaminase 2 (see e.g. Hohsfield LA et al., *Mol Cell Neurosci* 2014; 63:83-95, doi: 10.1016/j.mcn.2014.10.006).

-Figure 1a – the authors should clarify what they mean by reference angiogram.

-Results, line 190: observed vascular degenerations preceded the cognitive decline. This had already been shown earlier using vascular casting (see e.g. Kelly P et al., *Angiogenesis* 2017; 20(4):567-580, doi: 10.1007/s10456-017-9568-3). This should be discussed, pointing towards the 3Rs advantage of the present OCT method in providing in vivo information rather than relying on scanning electron microscopy examination of terminal vascular casts.

-Discussion (lines 224-227): The authors briefly mention the limitation that all imaging experiments were performed under anesthesia. The effect of anesthesia should be discussed more carefully. In OCT imaging of rodent brains, isoflurane may dilate cerebral vessels and inconsistent depths of anesthesia may introduce variability in vasoreactivity between imaging session. Anesthesia may have different effects in young and old animals, it may introduce a confounding effect in aging studies. Moreover, they mention follow-up studies in awake animals already taking place. Some of the results in awake animals should be presented, at least in WT mice, to verify whether anesthesia caused systematic effects in the assessments. This would enhance even more the value of the paper.

-Methods, line 368: Possible reasons for degrading image quality should be briefly mentioned. Were phase instabilities/errors the main reason?

-Craniotomy, lines 383-384: More details about the proprietary metal frame, used to hold the head

and prevent motion during microscopy, should be presented.

-OCT imaging (lines 409-427): Did any phase errors induced by sweep-trigger desynchronization need to be reduced (by e.g. spectral phase encoding and instantaneous correlation among the A-scans)? The authors used a NIR light source of 1.3 microns. Wouldn't it be more appropriate to use a longer wavelength in order to have a more homogeneous OCT depth penetration and to reduce scattering (e.g. 1.7 microns as shown by Chong SP et al., *Opt Lett* 2015;40(21):4911-4914, doi 10.1364/OL.40.004911)?

-OCT imaging: have the authors tested the sensitivity of the approach to detect vessel reactivity (e.g. following injection of e.g. acetazolamide)?

-In the Discussion section, a brief comparison between the present OCT method and more global assessments using diffuse correlation spectroscopy (e.g. Brothers RO et al., *Neurophotonics* 2021;8(1):015007, doi: 10.1117/1.NPh.8.1.015007), especially in the context of assessments on awake animals, should be made.

-The 3xTg model has little cerebral amyloid angiopathy. In a future work, it would be interesting to compare the present results with acquisitions on other models displaying vascular amyloid pathology, like APP23 or triple transgenic Tg-SwDI mice.

-Pathophysiological links are found between hypertension and Alzheimer's disease (see e.g. Carnevale D et al., *High Blood Press Cardiovasc Prev* 2016; 23(1):3-7, doi: 10.1007/s40292-015-0108-1). The data summarized in figures 4e,f and consistent with decreased arteriolar and venular diameters in 3xTg compared to WT mice (Table 1) suggest increased blood pressure with age. This should be properly discussed. Moreover, have peripheral blood pressure assessments using e.g. a noninvasive tail cuff been performed in the mice?

-Table 1: decreases in arteriolar diameter and flow were detected for both WT and 3xTg mice with age, although the effect was more pronounced for 3xTg animals. Possible consequences of this finding should be discussed.

-In a recent publication, Quintana et al. (*Neurobiol Aging* 2021; 105:115-128, doi: 10.1016/j.neurobiolaging.2021.04.019) showed microvascular degeneration occurring before plaque onset in 3xTg mice. In the present work, it would be interesting to compare the vascular changes with expected plaque load (obtained from other studies or even from the literature). Comparison to that work, showing hippocampal and cortical subregion-dependent changes to microvessels with age in 3xTg AD mice, as well as tortuosity changes, is warranted.

-The number and diameter of pial collateral "arterioles" decrease with aging is associated with reduced eNOS and increased oxidative stress. Mouse models of AD promote rarefaction of pial collaterals, suggesting inflammation-induced accelerated aging of collateral wall cells (e.g. Zhang H., *Angiogenesis* 2019; 22(2):263-279, doi: 10.1007/s10456-018-9655-0). Despite no difference in pial vessel diameter observed here between WT and 3xTg mice (Fig. 1), has there a difference in pial vessel number been detected? Information about changes in vessel density over the experimental period should be provided.

Minor comments:

-Introduction, line 44: should be "... OCT can rapidly produce microangiograms²¹, ... capillary vessels²³, all in a label-free manner."

-Results, line 88: Please specify whether 107 pial vessels were assessed for the different mice measured longitudinally.

-Results, line 101: Please specify whether 151 penetrating vessels were assessed for the different mice measured longitudinally.

-Results: please specify the meaning of the abbreviation WOA (weeks of age) the first time it

appears in the text.

-Line 226: should be "... awake imaging as being shown in our follow-up studies ...".

-Methods, Animal preparation, lines 365-368: the exact age of the mice at window placement should be provided.

-Line 418: should be "... 147,000 A-scans/s.".

Nicolau Beckmann

Reviewer #2 (Remarks to the Author):

This study provided some interesting advance in age-related alterations in the morphology, topology, and function of the cortical vasculature across all scales, which may be used to monitor the degenerative process of cerebral microvascular disease, especially, in Alzheimer's disease. However, there are some major problems in the reliability of technology, the theory proposed by the authors and statistical methods. This manuscript should be rejected or major revised before publication.

The problems are listed for reference:

1. Technical reliability

The experimental design had some shortcomings. For example, although the long-term longitudinal observation of vascular degeneration had been achieved by OCT, the inter-day and intra-day variations in this method should be tested in each mouse and provided in this study. Our comments mainly ensued from that these parameters of angiography are most likely affected by eating and drinking, room temperature or stress. Notably, the data in Figure 2c, 2f showed too much dispersion in WT group. Such discrete data are difficult or impossible to provide recognizable differences in actual imaging for diagnosing disease.

2. Theoretical reliability

The conclusion of temporal relationships between microvascular degenerations and cognitive impairment needs more sufficient evidences.

Although the authors found some differences in microvascular parameters between wild-type and AD model mice, this manuscript did not provide the convincing evidence, but only a novel object recognition experiment with no valid correlation analysis. The experiments of Morris water maze and fear conditioning test should be provided, and the relationship between the parameters of microvascular degeneration and the degree of cognitive decline should be investigated.

The minor problems:

1. Please provide details of the method of surgery for the chronic cranial window, including the specific location of the cranial window. How to make sure that the cranial window does not heal during the long experiment, thereby ensuring good imaging quality, should be explained or discussed it.

2. In Figure 1b, an example of selecting the same vessel across time points needs to be provided. Please explain: How to ensure that the same position of the same blood vessel was selected at different time points? Whether the diameter of blood vessels changes during the course of running or not? Why only a single cross-sectional diameter was used to represent the thickness of the blood vessel?

3. Please provide the acquisition time of all mice at each time point.

4. The results of the trans-genetic mice should be normalized to that of WT mice at each time point, so as to eliminate the influence of instruments and measurement conditions.

5. The manuscript mainly studied image registration and deep learning, etc. Please supplement the

physiological significance of the research.

6. Figure 1 displayed an example set of OCT angiograms to demonstrate that the technique is stable enough for longitudinal imaging. It will be more convincing if giving statistical results of quantitative parameters, such as the average intensity of one certain vessel at different time.

7. The left figure in Figure 4a could be displayed in different colors, similar to the right figure, to provide better visual effect.

8. In Discussion section, it mentioned that the reason why the structural degenerations precede the flow counterparts might be the effect of a compensating adaptation. What's the specific compensation adaptation conjecture? Also, please give a possible explanation for the morphological parameters on the physiological functions, similar to that of topology parameters in this manuscript.

9. Please give the loss or accuracy curve about the CNN network used to predict RBC flux in supplementary information.

10. Is it reasonable to use the TPM time traces to determine the ground truth? And when using it, how to solve the problem that the two-photon microscope will be used for long-term acquisition of red blood cell channel data? Is necessary to consider the possible problems of long-term acquisition.

11. When training the CNN model, explain the ratio of the training set and the validation set.

12. Is it okay to calculate the mean curvature of the vessels using the mean derivative of the tangent along the vessel without considering the influence of the thickness of the blood vessel?

13. For each figure and result, the specific number of mice and the number of repetitions to measure the selected vessel should be given.

14. In line 516, the author defined points whose Cook's distance was greater than three times the mean value as outliers and eliminates them. Generally, the median is selected as the threshold.

15. In line 513, the author used the AIC value to select the optimal model, it should be further explained by combining the ANOVA analysis results.

16. In line 529, the author determined the age at which AD and WT groups differed by assessing the degree of overlap between two confidence intervals. How to quantitatively measure the degree of overlap?

Response to Review

Manuscript ID: NCOMMS-22-09196-T

Reviewer 1

In the present work, Walek et al. longitudinally applied and combined multiple optical coherence tomography (OCT) techniques and integrated them with image processing algorithms to examine the microvasculature at different scales in wildtype and 3xTg mice modelling Alzheimer's disease. The elegant approach enabled longitudinal examination between 10 and 38 weeks, with imaging performed every 4 weeks.

We appreciate the reviewer for their careful reading of our manuscript and providing valuable feedback to enhance its quality. We have taken into consideration all of the comments and have conducted three additional experiments to address them in a data-supported manner. As a result, we believe that our manuscript has been significantly improved.

Main comments:

-More details about the simulation concerning the comparison between multi-endpoint and longitudinal approaches presented in suppl. Figure 1 should be provided.

We have described the simulation in more detail in the caption of Supplementary Fig. 1.

Supplementary information, line 12:

Supplementary Figure 1. An example of the multi-endpoint vs. longitudinal approach. These simulated data demonstrate how identical mean and standard deviations can result in different p-values. The illustrative data were simulated using a simple random number generation in MATLAB. Specifically, we set the means to be 1.0 and 1.5 for the groups 1 and 2, respectively, and then generated random numbers around the mean for each group, where the random numbers have a standard deviation of 0.5 for both groups and follow the normal distribution. We obtained the p-values via two-sample t-test (left) and paired t-test (right). In using the paired t-test, we assumed that the two groups represent two different time points in a longitudinal measurement.

-Introduction, lines 48-49: Important to mention is also the impact of repeated assessments on the same animal, especially of the chronic cranial window on aging animals.

We have added the following sentence to the Introduction to acknowledge this aspect.

Introduction, line 49:

However, it is unclear for how long OCT methods can repeatedly and robustly image the same cortical microvasculature, and the impact of aging on this imaging. In addition, it is also important to consider the impact of repeated assessments on the same animal, particularly the potential effect of the chronic cranial window on aging animals.

-Methods, Animal preparation, lines 365-368 & 403-405: the authors state that installing a chronic cranial window in mice younger than 10-12 weeks can be problematic due to incomplete skull growth prior to this age. Assuming that the mice were 10 weeks old at the beginning of the study, and taking the 7-month-observation period, would lead to an age of approximately 280 days at the end of the study. For male 3xTg mice, the survival rate has been shown to be 80% at this age, while for WT mice, it is almost 100% (Kane AE et al., Front Aging Neurosci 2018; 10:172, doi: 10.3389/fnagi.2018.00172). In the present work, 30%-40% of the mice including wildtype mice, died during the longitudinal assessments over the course of 7 months or were excluded due to image degradation. This seems a quite high number, clearly suggesting to me that the window placement might have a long term influence on the physiology of the animals. Histological analyses of the brain of mice at the age of 280 days should be provided, comparing 3xTg and WT mice with and without cranial window, addressing possible side effects of the window.

We appreciate the reviewer's insightful suggestion and constructive criticism. We apologize for the lack of detail in our original manuscript and appreciate the opportunity to clarify the point.

As described in lines 398-403 of the original manuscript, four animals did not survive until the end of the 7-month experiment. Three of these animals, however, did not die spontaneously but were euthanized by us due to mechanical damage to their window headposts. The damage occurred while they were moving in their cages. Since this euthanasia was performed to meet the humane endpoints of our IACUC, the death concerned by the reviewer occurred only in one mouse out of 20 mice, which is a slightly lower percentage than the one cited by the reviewer. Therefore, the survival rate alone does not necessarily suggest that the window placement might have a long-term adverse effect on the physiology of the animals.

To further address the potential long-term adverse effects of the chronic cranial window on the physiology of the animals, we conducted an additional experiment (**Revision Experiment A**) as suggested by the reviewer. We fortunately had collected and fixed the brains of the animals after the longitudinal experiment. Here we fixed, sectioned, and stained these brains. We then performed

immunofluorescence imaging to investigate the possible side effects of the window. We used the widely used GFAP and IBA-1 stains to visualize astrocyte and microglia immunoreactivity, respectively. The immunoreactivity was measured by the ratio of GFAP to DAPI or the ratio of IBA-1 to DAPI in either cell number or pixel area. We found no statistically significant differences in the immunoreactivity between the ipsilateral and contralateral cortices of the window installation, for both AD and WT mice.

Our new Supplementary Text 8 provides a detailed description of this additional experiment and its results. As discussed in the Supplementary Text, although the literature suggests that craniotomy may lead to higher immunoreactivity within 3-4 weeks after installation, this immunoreactivity typically returns to normal thereafter. This consistent finding is supported by our results. We observed no difference in astrocyte or microglia immunoreactivity between the ipsilateral and contralateral cortices of the window installation, even though the animals had undergone longitudinal imaging for seven months.

In conclusion, the additional experiment we conducted, along with the literature, suggests that a chronic cranial window may not have significant long-term adverse effects on the physiology of the animals. Moreover, it is important to note that even if a possible adverse effect of long-term craniotomy existed, but was not detected, it would not produce a significant difference in the major outcome of the presented study, as we compared relative changes with age (the rate of change with age, RCA) rather than absolute values of vascular properties between AD and WT mice. Therefore, it is unlikely that the possible long-term adverse effect of the window placement on cortical physiology would affect the conclusions of our study.

We have added the new Supplementary Text 8 and modified the text in the manuscript to reflect these findings as follows. We hope that these clarifications and additional experiments have addressed the reviewer's concerns, and we are grateful for their helpful comments.

Methods, line 416:

Of these mice, one mouse died during the experiment, three mice were euthanized due to damage on their headposts, and three mice were excluded due to degrading image quality (see Supplementary Text 8 for potential impact of long-term craniotomy and imaging on animal physiology).

Methods, line 467:

... we began with 20 animals, but one animal did not survive until the end of our seven-month longitudinal experiment. In three out of the surviving animals, the imaging quality through the window degraded significantly with time, secondary to scar tissue formation underneath the window such that insufficient number of vessels were visible with imaging - these animals were not included in our analysis. Additionally, three animals were euthanized in the middle of experiment due to headpost damage.

Supplementary information, line 424, new section:

Supplementary Text 8. Potential impact of long-term craniotomy and imaging on animal physiology.

As described in the Methods section, we started with 20 animals, but one animal did not survive until the end of our seven-month longitudinal experiment. Three animals were euthanized in the middle of the study due to mechanical damage to their headposts, and three animals were excluded due to degrading image quality. The degrading image quality (3 out of 20) was first noticeable as blurry capillaries in microangiograms, likely due to the formation of a thin layer of scar tissue below the bottom surface of the glass and the top surface of the cortex. The mechanical damage (3 out of 20) occurred while the animals were moving in their cages. The spontaneous death rate (1 out of 20, 5%) was comparable to the known survival rate at the corresponding age⁴⁰. This consistent survival rate suggests that chronic cranial windows might not have a severe long-term influence on the physiology of animals.

Several studies have investigated the effect of open-skull craniotomy on cortical physiology, particularly in terms of immunoreactivity. The literature suggests that the craniotomy may lead to higher immunoreactivity within 3-4 weeks after installation, but the immunoreactivity returns to normal after that time. For example, Xu et al. reported higher immunoreactivity in the window-installed cortex than the other side within 20 days of installation, but no difference at 30 days⁴¹. Holtmaat et al. also reported that astrocytes, not microglia, became more immunopositive in the window-installed cortex at 2 weeks but returned to normal at 4 weeks of window installation⁴². Similarly, Goldey et al. reported no difference between the ipsilateral and contralateral cortices of window installation at about 10 days⁴³. Heo et al. found that a PDMS window installation led to a high microglia cell density, not astrocytes, at 1 week but the density returned to normal at 3 weeks⁴⁴.

Although the spontaneous death rate and existing literature suggest that long-term craniotomy is unlikely to have adverse effects on animal physiology, there has been no direct investigation into this possibility. To address this, we conducted post-mortem immunofluorescence imaging of the brains of animals that had undergone our longitudinal experiment. Using the published method of Heo et al.⁴⁴, we fixed, sectioned, and stained the brains with GFAP and DAPI (Supplementary Fig. 9a) or IBA-1 and DAPI (Supplementary Fig. 9d), which respectively visualize astrocyte and microglia immunoreactivity. These two stains are widely used in the literature to investigate the effect of craniotomy on cortical physiology⁴¹⁻⁴⁴. We measured astrocyte immunoreactivity by calculating the ratio of GFAP to DAPI in terms of cell number or pixel area, and found no statistically significant differences between the ipsilateral and contralateral cortices of the window installation in both ratios (Supplementary Fig. 9c). We performed similar analysis on the DAPI and IBA-1 stained images and found no statistically significant differences in microglia immunoreactivity (Supplementary Fig. 9f).

Supplementary information, line 92, new figure:

Supplementary Figure 9. Post-mortem immunofluorescence result. **a)** An example of GFAP and DAPI stain images. The yellow polygons indicate the regions of interest (ROIs) used for quantitative analysis in (c). Scale bar, 1 mm. **b)** Magnified images of the white rectangular areas in (a). Scale bar, 0.1 mm. **c)** 17 slices were analyzed (12 slices from 3 AD mice and 5 slices from 2 WT mice). All GFAP images underwent an identical image segmentation process, which used adaptive thresholding and filtered foreground objects by the area (equivalent to circles of 10-30 μm in diameter). DAPI images underwent a circle detection processing. The differences were normally distributed, and we used paired t-tests to analyze them. We excluded one slice as an outlier in the area ratio analysis. The effect of AD on this conclusion of insignificant difference was not statistically significant ($p=0.193$, number ratio; 0.267 , area ratio; linear mixed-effect [LME] analysis). **d)** An example of IBA-1 and DAPI stain images. The yellow polygons indicate the ROIs used for quantitative analysis in (f). Scale bar, 1 mm. **e)** Magnified images of the white rectangular areas in (d). The IBA-1 channel images underwent contrast adjustment to suppress background fluorescence. **f)** 13 slices were analyzed (9 slices from 2 AD and 4 slices from 1 WT mice). All IBA-1 images underwent the identical image segmentation as that for GFAP images. DAPI images also underwent the identical circle detection processing. The differences were normally distributed, and we

used paired t-tests to analyze them. The effect of AD on this conclusion of insignificant difference was not statistically significant ($p=0.093$, number ratio; 0.054 , area ratio; LME analysis).

-It would be quite interesting to compare the results on OCT microangiography obtained here with a more global technique, e.g. perfusion MRI or transcranial Doppler ultrasound. On the other hand, for validation purposes it would have been important to compare the OCT results with histological analysis of microvasculature, e.g. using collagen IV and transglutaminase 2 (see e.g. Hofsfield LA et al., Mol Cell Neurosci 2014; 63:83-95, doi: 10.1016/j.mcn.2014.10.006).

Thanks to this insightful suggestion, we have extended the scope of comparison beyond in-vivo microscopic imaging results. Although no studies to our knowledge tracked cerebrovascular alterations over a long period of time comparable to ours, which makes the extended comparison often difficult, we have added discussion to the Discussion section and Supplementary Text 7.

Discussion, line 261, newly added:

Although the primary focus of our study is technical demonstration rather than biological discovery (Supplementary Text 4), the observed cortical hypoperfusion in AD is consistent with findings from perfusion magnetic resonance imaging and transcranial Doppler ultrasound studies, which have reported hypoperfusion in various cortical regions in early human AD (see Refs. 33 and 34 for reviews).

Discussion, line 272:

The mean capillary length result (becoming shorter with aging in AD only, Fig. 3d) is consistent with the previous finding that capillary length is shorter in AD than WT²³, although the previous study did not longitudinally track the length and used a different AD mouse model (see Supplementary Text 7 for further discussion).

Supplementary information, line 368, new paragraph:

Our results showing a decrease in mean capillary length with aging in AD (Fig. 3d) are consistent with a previous study that found capillary length to be shorter in AD mice than in wild-type (WT) mice at 18-31 WOA⁷, although that study did not longitudinally track the length and used a different AD mouse model. Another study found that capillary branch numbers were lower in 3xTg mice than WT mice at 20 months of age, but no difference was observed at 7 and 14 months²⁶. This finding is consistent with our result of shorter capillary lengths in 3xTg mice, but our measurement revealed the difference much earlier (Fig. 3d). There are several possible reasons for this discrepancy, including that (i) the statistical analysis in the previous study did not consider the clustered nature of the data, leading to lower sensitivity to intra-group differences; (ii) our method tracks the trend of the capillary property varying with age, whereas their method relies on a

snapshot at a specific age; and/or (iii) our method analyzes 3D networks, while their method analyzes only 2D cross-sections of the network.

-Figure 1a – the authors should clarify what they mean by reference angiogram.

We have clarified it in the caption of Fig. 1 of the revised manuscript.

Figures, line 321:

a) An example of registration of an angiogram (cyan) to a reference angiogram (red) with overlapping regions shown in white. To achieve this, we selected one angiogram as a reference out of the seven angiograms and used our code to shift and rotate the other six to align their imaging area and angle with the reference. This registration enabled us to visually identify and mark the same vessels across all seven angiograms.

-Results, line 190: observed vascular degenerations preceded the cognitive decline. This had already been shown earlier using vascular casting (see e.g. Kelly P et al., *Angiogenesis* 2017; 20(4):567-580, doi: 10.1007/s10456-017-9568-3). This should be discussed, pointing towards the 3Rs advantage of the present OCT method in providing in vivo information rather than relying on scanning electron microscopy examination of terminal vascular casts.

Thanks to this suggestion, we have improved the relevant part of Discussion.

Discussion, line 221, newly added:

Moreover, the presented methods align with the “3 R’s” principles of animal research, particularly reduction. For example, while a vascular corrosion casting study demonstrated age-related changes in capillary morphology, it required more animals due to the terminal nature of the method⁵. Similarly, another casting study identified that changes in capillary morphology preceded cognitive decline, but it was limited to investigating only one age point out of a similar number of animals³¹. In contrast, the in vivo nature of the OCT method used in this study provides a clear advantage in reducing the number of animals required, while simultaneously providing detailed information about the dynamics of cerebrovascular changes.

-Discussion (lines 224-227): The authors briefly mention the limitation that all imaging experiments were performed under anesthesia. The effect of anesthesia should be discussed more carefully. In OCT imaging of rodent brains, isoflurane may dilate cerebral vessels and inconsistent depths of anesthesia may introduce variability in vasoreactivity between imaging session. Anesthesia may have different effects in young and old animals, it may introduce a confounding effect in aging studies. Moreover, they mention follow-up studies in awake

animals already taking place. Some of the results in awake animals should be presented, at least in WT mice, to verify whether anesthesia caused systematic effects in the assessments. This would enhance even more the value of the paper.

Thanks to this insightful comment, we have conducted an additional experiment (**Revision Experiment B**) to directly compare the vascular effect of isoflurane anesthesia between young and old mice. In this experiment, we used a separate cohort of young (15 WOA) and old (55 WOA) WT mice (C57BL/6J, n=4 each) and performed OCT imaging under two different anesthetic conditions: 1.5% isoflurane (with oxygen flow of 1 L/min) and 100% oxygen (without anesthesia). We imaged the same cortical region twice between the two imaging sessions. We analyzed the changes between the two imaging sessions in all 25 vascular properties as in the main study. We have added Supplemental Text 9 to describe this experiment and its result, and accordingly have improved the related part of the Discussion section.

Discussion, line 237:

One limitation in the presented demonstration is that we conducted all imaging experiments under anesthesia, although anesthesia unlikely affects our major outcome, i.e., difference in the rate of change with age (RCA) between AD and WT (see Supplementary Text 9 for related data and discussion). The presented methods are readily applicable to awake imaging as being shown in our follow-up studies.

Supplementary information, line 464, new section:

Supplementary Text 9. Impact of anesthesia on OCT measurements of vascular morphology, topology, and flow.

The presented study conducted OCT imaging under isoflurane anesthesia, which is known to dilate cerebral vessels. While this vasodilation effect may vary between imaging sessions, it would work as random noise and not affect the main conclusion of the study. This is because the study focused on comparing relative changes in the rate of change with age (RCA) between AD and WT groups, rather than absolute values. However, if the vasodilation effect is systematically different between younger and older mice, it could affect the accuracy of the RCA measurement. Even in this case, it would not change the main conclusion of the study, as the focus was on the statistical difference in RCA between the groups, rather than absolute RCA values.

Nonetheless, it would be valuable to identify which vascular properties are affected by isoflurane anesthesia in an age-related manner, and which are not. As no study has comprehensively investigated this for younger and older mice, an additional experiment was conducted in WT mice.

Methods: Seven days after installing a chronic cranial window on a mouse as described in the main text, we placed the mouse on an air-floating platform (Mobile Homeage, Neurotar) for a head-fixed, freely-walking awake imaging condition, while connecting the mouse nose to the isoflurane vaporizer. Under this awake condition, we acquired a similar set of OCT data as in the main study, including OCT angiogram, Doppler OCT,

microangiogram, and RBC passage data. Then, we turned on the isoflurane supply (1.5% isoflurane with oxygen flow of 1 L/min), and after 30 minutes, we repeated the OCT data acquisition under this anesthetized condition. We repeated this experiment in young (15 WOA) and old (55 WOA) mice (n=4 mice per age group). This dataset underwent the same analysis as described in the main study to measure all the vascular properties listed in Table 1. For statistical analysis of properties tracked vessel by vessel (Supplementary Fig. 9), we used the LME method involving animal-specific random effects, excluding any outliers when a residual was more than three scaled median absolute deviations from the median. When this LME method resulted in non-normally distributed residuals, we used a bootstrapped LME method with a bootstrapping number of 2,000. Since this bootstrapped LME analysis, like other bootstrap methods, only provides confidence intervals (CIs), we considered it statistically significant ($p < 0.05$) when the bootstrapped CI with $\alpha = 0.05$ does not involve zero. In the statistical analysis of capillary properties, where we averaged values of a property over all capillaries within each animal, since we cannot track thousands of the same capillaries, the sample size was small (4 animals per age group), so we used the bootstrap method to find the CI from 1,000 bootstraps.

Results: As summarized in Supplementary Table 3, isoflurane anesthesia at the concentration we used statistically significantly increased the pial vessel diameter, arteriolar diameter, venular diameter, arteriolar flow, capillary diameter, capillary diameter COV, capillary RBC flux, and capillary RBC flux COV in young mice. It was interesting to confirm that significant changes were observed only for those properties expected to be affected by the isoflurane-induced vasodilation and hyperperfusion, except for venular flow ($p = 0.092$), whereas other structural properties did not exhibit statistically significant changes, as expected, including capillary length, tortuosity, betweenness, and closeness. This distinct result among capillary morphological, topological, and flow properties supports the rigor of our approach.

Returning to the focus of this additional experiment, whether the isoflurane effect is different between young and old mice, the effect size of the older age was statistically significant for pial vessel diameter only (Supplementary Table 3, Supplementary Fig. 9a). The isoflurane-induced increase in pial vessel diameter was smaller in old mice ($p < 0.05$).

Discussion: The isoflurane-induced vasodilation effect on pial vessel diameter was greater in younger mice. Therefore, when tracking the pial vessel diameter with aging and measuring its slope (RCA) as we did in the main study, a negative RCA may be observed even if the animals do not exhibit physiological decreases in pial vessel diameter. As expected, our main study observed a small negative RCA in pial vessel diameter, with no statistically significant difference between AD and WT mice (-1.3%/month and -1.1%/month for AD and WT, respectively, $p = 0.78$ between AD and WT, Table 1). This negative RCA is highly likely due to the age-dependent pial vessel dilation effect of isoflurane. The effect size of the older age was -15% on average (Supplementary Table 3) across approximately 10 months of age (between 15 WOA and 55 WOA). This effect size is expected to generate an artifact RCA of -1.5%/month, assuming that the age-dependent pial vessel dilation effect can be linearly interpolated

between 15 and 55 WOA. However, this artifact RCA falls within the confidence intervals of the pial vessel diameter RCAs of both AD and WT mice as measured in the main study (-2.4 to -0.2 and -2.2 to 0.0 %/month in AD and WT, respectively, Table 1).

Supplementary information, line 114, new figure:

Supplementary Figure 10. Effects of isoflurane anesthesia on individual vessels. **a)** Effects of isoflurane on pial vessel diameter. 61 vessels (young) and 59 vessels (old) were measured and analyzed by bootstrapped LME, as traditional LME analysis produced non-normally distributed residuals. **b)** Effects of isoflurane on arteriolar and venular diameter and flow. 11 arterioles and 15 venules (young) and 16 arterioles and 17 venules (old) were measured and analyzed by LME. Residuals were normally distributed.

Supplementary information, line 154:

Supplementary Table 3. Effects of isoflurane anesthesia on vascular properties. Statistically significant ($p < 0.05$) values are presented in bold. The property name with a statistically significant difference in isoflurane effect between young and old mice is presented in bold.

Properties	Relative change (% , 95% CI) in young mice	Effect size of the older age (percent points, 95% CI) and its p-value	Sample size (young and old) and the used method
Pial vessel diameter	24 to 35	-24 to -6 ($p < 0.05$)	61 and 59 vessels, bootstrapped LME
Arteriolar diameter	7 to 27	-12 to 14 ($p = 0.878$)	11 and 16 vessels, LME

Venular diameter	3 to 50	-41 to 25 (p=0.608)	15 and 17 vessels, LME
Penetrating vessel density	-16 to 13	-19 to 22 (p = 0.893)	4 animals per group, bootstrap
Arteriolar flow	24 to 108	-86 to 31 (p=0.335)	11 and 16 vessels, LME
Venular flow	-5 to 60	-47 to 43 (p=0.923)	15 and 17 vessels, LME
Capillary length*	-8 to -2	-1 to 19 (p=0.341)	4 animals per group, bootstrap
Capillary length COV*	-9 to 4	-7 to 15 (p=0.522)	4 animals per group, bootstrap
Capillary diameter*	1 to 23	-18 to 14 (p=0.956)	4 animals per group, bootstrap
Capillary diameter COV*	7 to 25	-21 to 16 (p=0.483)	4 animals per group, bootstrap
Capillary tortuosity*	-3 to 5	-3 to 10 (p=0.422)	4 animals per group, bootstrap
Capillary tortuosity COV*	-34 to 55	-36 to 120 (p=0.336)	4 animals per group, bootstrap
Branching order*	-13 to 18	-20 to 23 (p = 0.854)	4 animals per group, bootstrap
Branching order COV*	-15 to 35	-49 to 21 (p = 0.413)	4 animals per group, bootstrap
Betweenness*	-24 to 30	-89.5 to -0.4 (p=0.184)	4 animals per group, bootstrap
Betweenness COV*	-21 to 21	-21 to 69 (p=0.426)	4 animals per group, bootstrap
Closeness*	-19 to 127	-104 to 209 (p=0.883)	4 animals per group, bootstrap
Closeness COV*	-3 to 20	-15 to 34 (p=0.785)	4 animals per group, bootstrap
Shortest cycle*	-6 to -1	-3 to 4 (p=0.839)	4 animals per group, bootstrap
Shortest cycle COV*	-3 to 11	-19 to 12 (p=0.807)	4 animals per group, bootstrap
Capillary number density	-2 to 2	-7 to 11 (p=0.384)	4 animals per group, bootstrap

Capillary length density	-1 to 2	-6.0 to 0.3 (p=0.305)	4 animals per group, bootstrap
Fractal dimension	-0.2 to 7.5	-8 to 3 (p=0.921)	4 animals per group, bootstrap
RBC flux*	4 to 13	-5 to 6 (p=0.563)	4 animals per group, bootstrap
RBC flux COV*	-23 to -8	-22 to 3 (p=0.326)	4 animals per group, bootstrap

* These capillary properties were measured and averaged (or calculated for COV) over 407-1718 capillaries within each animal (min-max) prior to being compared between the age groups.

-Methods, line 368: Possible reasons for degrading image quality should be briefly mentioned. Were phase instabilities/errors the main reason?

We have added this point to Supplementary Text 8, where we discussed various impacts of the chronic cranial window installation and described the new immunofluorescence imaging result. The image quality was gradually degraded throughout a long period of time, only in 3 out of the 20 animals, likely due to biological reasons rather than technical reasons like phase instability.

Methods, line 418:

... were excluded due to degrading image quality (see Supplementary Text 8 for potential impact of long-term craniotomy and imaging on animal physiology).

Supplementary information, line 427:

As described in the Methods section, we started with 20 animals, but one animal did not survive until the end of our seven-month longitudinal experiment. Three animals were euthanized in the middle of the study due to mechanical damage to their headposts, and three animals were excluded due to degrading image quality. The degrading image quality (3 out of 20) was first noticeable as blurry capillaries in microangiograms, likely due to the formation of a thin layer of scar tissue below the bottom surface of the glass and the top surface of the cortex. The mechanical damage (3 out of 20) occurred while the animals were moving in their cages.

-Craniotomy, lines 383-384: More details about the proprietary metal frame, used to hold the head and prevent motion during microscopy, should be presented.

We have added Supplementary Fig. 11 for details about the headpost.

Methods, line 437:

A proprietary metal frame, used to hold the head and prevent motion during microscopy (Supplementary Fig. 11), was affixed to the skull with dental cement (Parkell Inc., Long Island, NY, USA).

Supplementary information, line 124, new figure:

Supplementary Figure 11. Metal frame for head fixation during imaging. (Left) A picture of the metal frame used to affix the skull during the craniotomy surgery. The frame holds the head and minimizes motion artifacts during OCT imaging. (Right) The dimensions of the metal frame used in this study.

-OCT imaging (lines 409-427): Did any phase errors induced by sweep-trigger desynchronization need to be reduced (by e.g. spectral phase encoding and instantaneous correlation among the A-scans)? The authors used a NIR light source of 1.3 microns. Wouldn't it be more appropriate to use a longer wavelength in order to have a more homogeneous OCT depth penetration and to reduce scattering (e.g. 1.7 microns as shown by Chong SP et al., *Opt Lett* 2015;40(21):4911-4914, doi 10.1364/OL.40.004911)?

We used spectral-domain OCT, not swept-source OCT, and thus were free from the specific type of phase errors that are induced by sweep-trigger desynchronization. Regarding the wavelength, we used 1.3 μm as it is long enough to cover all cortical depths and the scope of this study focuses on cortical microvasculature. But we agree that the use of a longer wavelength like 1.7 μm would enable further investigation of subcortical vasculature. We have added this point to the related part of Supplementary Text 3.

Supplementary information, line 252, new paragraph:

Another potential opportunity of the presented method is to investigate subcortical vasculature when being used with a longer-wavelength OCT. We used the OCT with the center wavelength of 1.3 μm as it covers all cortical depths and the scope of this study focuses on cortical microvasculature. OCT with longer wavelengths like 1.7 μm , however, can provide more homogeneous depth penetration and potentially reduce scattering, enabling further investigation of subcortical vasculature¹⁴. Combining the presented method with such a longer-wavelength OCT could enable researchers to explore how the vasculature beneath the cortex gradually alters in animal models of Alzheimer's disease, and how the alteration differs from that of cortical vasculature.

-OCT imaging: have the authors tested the sensitivity of the approach to detect vessel reactivity (e.g. following injection of e.g. acetazolamide)?

We have conducted an additional experiment to directly quantify the sensitivity of our approach in detecting changes in vessel diameter and blood flow (**Revision Experiment C**). We used isoflurane anesthesia, which is known to induce vasodilation in cerebral vasculature. The results of this additional experiment are presented in Supplementary Text 10 and Supplementary Fig. 12. The data show that our approach is sensitive enough to detect changes in vessel diameter and flow, with a conservative detection limit of about 7% for pial vessel diameter, about 6% for arteriolar blood flow, and about 24% for arteriolar blood flow. Although the statistical results presented in the original manuscript support that our approach was sensitive enough to reveal subtle differences in those vascular changes with age between AD and WT mice, this additional data of directly quantifying the measurement sensitivity further strengthens the rigor of our approach.

Discussion, line 246:

Supplementary Text 3 provides further discussion of other challenges and opportunities related to the presented methods. Supplementary Text 10 discusses the sensitivity of our approach to detecting changes in vascular diameter and blood flow.

Supplementary information, line 532, new section:

Supplementary Text 10. Sensitivity of OCT measurements of changes in vascular diameter and blood flow.

Although the statistical results presented in the main text support that our approach was sensitive enough to reveal subtle differences in those vascular changes with age between AD and WT mice, we conducted a separate experiment to directly quantify the sensitivity of our approach of detecting changes in the average diameter and blood flow of pial vessels and penetrating arterioles.

Methods: We used isoflurane as a vasodilator and repeated OCT imaging while vessels went through dynamic changes via the dilation and recovery. In detail, seven days after we installed a chronic cranial window on a mouse as described in the main text, we put the mouse on an air-floating platform (Mobile Homecage, Neurotar) for a head-fixed, freely-walking awake imaging condition, while the mouse nose is connected to the isoflurane vaporizer. We acquired nine sets of OCT angiogram and Doppler OCT images, following the exact protocols described in the main text, at the following time points:

1. Awake as a baseline
2. 5 minutes after turning ON isoflurane supply
3. 25 minutes after turning ON isoflurane supply
4. Right after turning OFF isoflurane supply
5. 7 minutes after turning OFF isoflurane supply
6. 15 minutes after turning OFF isoflurane supply

7. 30 minutes after turning OFF isoflurane supply
8. 45 minutes after turning OFF isoflurane supply
9. 60 minutes after turning OFF isoflurane supply

This longitudinal dataset underwent the analysis process as described in the main text, in order to identify same vessels across time points and measure the pial vessel diameter, arteriolar diameter and flow, and venular diameter and flow, for each vessel and each time point. We repeated this data acquisition and analysis in four young WT mice. For statistical analysis of a change between two states, we used linear mixed-effect (LME) analysis with the nested random effects of individual vessels in animals. This nested LME was identical to the LME analysis used in the main text.

Results: As expected, the pial vessel diameter increased with isoflurane and returned to the baseline after recovery (Supplementary Fig. 12a). Our approach was sensitive enough to detect a detectable 7% change in the pial vessel diameter between the first awake state and the state 5 minutes after turning on the isoflurane supply. The average pial vessel diameter in the awake state was 19.7 μm , and the increased diameter in the second state was 1.4 μm ($p=0.026$, $n=59$ vessels from four animals; Supplementary Fig. 12b). Our approach was also robust enough to produce statistically insignificant results between the first and the final states as physiologically expected ($p=0.932$, Supplementary Fig. 12c) while still detecting the 1.4- μm change.

Similarly, the arteriolar diameter and flow increased with isoflurane and returned to the baseline after recovery, while the venular diameter and flow did not (Supplementary Fig. 12d). We found subtle fluctuations during the recovery phase, the degree of which our approach was able to detect. The average arteriolar diameter in the first of the final two states (45 minutes after turning off isoflurane supply) was 27.5 μm , and the increased diameter in the final state was 1.6 μm , a 6% change ($p=0.022$, $n=7$ arterioles from four animals; Supplementary Fig. 12e, top). The average arteriolar flow changed by 0.098 $\mu\text{L}/\text{min}$ from 0.409 $\mu\text{L}/\text{min}$, a 24% change ($p=0.052$, Supplementary Fig. 12e, bottom). Our approach was also robust in producing statistically insignificant results between the first and the final states as biologically expected ($p=0.942$ and 0.436 for diameter and flow, respectively; Supplementary Fig. 12f).

Discussion: The supplementary experiment presented here provides a quantitative measure of the sensitivity of our approach for detecting changes in vascular diameter and blood flow. We found the sensitivity of detecting changes in pial vessel diameter, arteriolar diameter, and arteriolar blood flow to be 1.4 μm (7% change), 1.6 μm (6% change), and 0.098 $\mu\text{L}/\text{min}$ (24% change), respectively. It is important to note that these sensitivity values serve as a conservative estimate of the actual sensitivity of the measurements described in the main text. This is because the supplementary experiment used fewer animals (four, against 6-7 per group of the main study) and compared only two time points, whereas the main experiment employed seven time points to determine the slope (rate of change with age, RCA). It is likely that the RCA measurement described in the main text has higher statistical power and thereby higher sensitivity.

Supplementary information, line 132, new figure:

Supplementary Figure 12. OCT-measured changes in vascular diameter and flow during vasodilation and recovery. **a)** Tracked vessel diameter over 59 same pial vessels in four animals, with the blue line indicating the simple average of the diameters for each time point. **b)** Pial vessel diameter changes between the awake state and the state after 5 minutes of turning on isoflurane supply. The selected time points produced a p value close to 0.05. **c)** Pial vessel diameter changes between the first state (awake) and the final state (after 60 minutes of turning off isoflurane supply). **d)** Tracked vessel diameter and flow over seven same arterioles and 13 same venules in four animals. **e)** Arteriolar diameter and flow changes between the state before the final state (45 minutes after turning off isoflurane supply) and the final state, with the selected time points producing p values close to 0.05. **f)** Arteriolar diameter and flow changes between the first and final states.

-In the Discussion section, a brief comparison between the present OCT method and more global assessments using diffuse correlation spectroscopy (e.g. Brothers RO et al., Neurophotonics 2021;8(1):015007, doi: 10.1117/1.NPh.8.1.015007), especially in the context of assessments on awake animals, should be made.

We have added a comparison to the Discussion section.

Discussion, line 240:

The presented methods are readily applicable to awake imaging as being done shown in our follow-up studies. In the context of longitudinal assessments of awake animals, our methods can offer detailed, microvasculature-level assessments under a head-fixed condition. In comparison, diffuse correlation spectroscopy has shown the ability to provide assessments under a freely-behaving condition but at a less detailed, macroscopic level³². Combining these two methods could yield a more comprehensive understanding of the changes and mechanisms underlying cerebrovascular dysregulation in Alzheimer's disease and other progressive vascular diseases. Supplementary Text 3 provides further discussion of other challenges and opportunities related to the presented methods.

-The 3xTg model has little cerebral amyloid angiopathy. In a future work, it would be interesting to compare the present results with acquisitions on other models displaying vascular amyloid pathology, like APP23 or triple transgenic Tg-SwDI mice.

We absolutely agree with the reviewer. As clarified in Supplementary Text 4 of the original manuscript, we chose the 3xTg model primarily as a vehicle for demonstrating the capability of our methods rather than for producing biological discovery. In future work, we will definitely consider studying other models that display vascular amyloid pathology, like APP23 and triple transgenic Tg-SwDI models. We have added the potential of future work to the Supplementary Text 4.

Supplementary information, line 317, new paragraph:

Lastly, it should be noted that the 3xTg model used in this study has little cerebral amyloid angiopathy, and thus, to specifically study vascular amyloid pathology, other models such as APP23 and triple transgenic Tg-SwDI models^{24,25} may be employed. It would be interesting to compare the results presented here to those obtained from other models in future work.

-Pathophysiological links are found between hypertension and Alzheimer's disease (see e.g. Carnevale D et al., High Blood Press Cardiovasc Prev 2016; 23(1):3-7, doi: 10.1007/s40292-015-0108-1). The data summarized in figures 4e,f and consistent with decreased arteriolar and venular diameters in 3xTg compared to WT mice (Table 1) suggest increased blood pressure

with age. This should be properly discussed. Moreover, have peripheral blood pressure assessments using e.g. a noninvasive tail cuff been performed in the mice?

We appreciate the reviewer's insightful comment. Hypertension, especially how blood pressure changes with age in AD and its relation to cerebral microvascular changes, is one of our future focuses of applying the presented approach in diverse AD models. We agree that our data suggest that the blood pressure might have increased in the AD animals used in our study, as indicated by the decreased arteriolar and venular diameters in 3xTg compared to WT mice (Table 1). However, we did not longitudinally measure blood pressure in our study. In future experiments, we plan to perform a long-term trace of blood pressure using a noninvasive method such as the tail-cuff technique in diverse AD models. This will enable us to compare the long-term blood pressure trace with the traces of cerebral microvascular properties obtained by the presented methods.

We have added this point to Supplementary Text 7. We also added a brief discussion on the link between hypertension and AD, and the challenges associated with investigating this link in human studies. We also included a reference to a recent systemic review that highlights the importance of investigating the interactions among plaques, tangles, cerebrovascular pathology, and dementia to better understand the role of hypertension in AD development. We hope these additions will strengthen the manuscript's discussion.

Discussion, line 284:

The presented results also showed degenerations in capillary blood flow (see Supplementary Text 7 for interpretation and potential relation to hypertension).

Supplementary information, line 411, new paragraph:

The decrease in penetrating arteriole and venule diameters (Table 1) further suggests that the blood pressure might have increased with age in the 3xTg mice we used. Unfortunately, we did not longitudinally measure blood pressure in our study. Hypertension has been linked to AD, and animal studies suggest that hypertension can lead to amyloid plaques, neuroinflammation, blood-brain barrier dysfunction, and cognitive impairment³⁷. However, human studies on the association between hypertension and AD have been sparse and inconsistent, and the effect of antihypertensive medications on AD seems weak^{38,39}. Investigating interactions among plaques, tangles, cerebrovascular pathology, and dementia may be key to understanding hypertension's role in AD development³⁹. Therefore, our future work includes performing a long-term trace of blood pressure using a noninvasive method like the tail-cuff technique in diverse AD models, which will enable us to compare the long-term blood pressure trace with the traces of cerebral microvascular properties obtained by the presented methods.

-Table 1: decreases in arteriolar diameter and flow were detected for both WT and 3xTg mice with age, although the effect was more pronounced for 3xTg animals. Possible consequences of this finding should be discussed.

We have added a discussion of this point to Supplementary Text 7.

Supplementary information, line 396, new paragraph:

The combined results of arteriolar diameter/flow and capillary flow suggest another possible consequence of decreased arteriolar diameter and flow, namely an increase in the presence of hypoxic micro-pockets in the cortex. The observed decreases in both arteriolar diameter and flow would lead to arteriolar and near-arteriolar tissue oxygen pressure (pO₂)³⁴. This impaired arteriolar oxygen delivery would not immediately lead to an impairment in capillary-bed tissue pO₂ when the compensatory mechanism by enhanced capillary flow works as hypothesized above from our observed capillary RBC flux increases. However, when the compensatory mechanism no longer works at later ages, the capillary-bed tissue pO₂ would become lower and spatially more heterogeneous, as observed between 60 and 100 WOA in WT mice³⁴. This impaired capillary oxygen supply increases the presence of hypoxic micro-pockets, which may explain the observation of tinier microinfarcts in aging brains, particularly those with mild cognitive decline or AD^{35,36}. Therefore, the decrease in arteriolar diameter and flow observed in both WT and 3xTg mice with age may have important implications for the development and progression of neurodegenerative diseases.

-In a recent publication, Quintana et al. (Neurobiol Aging 2021; 105:115-128, doi: 10.1016/j.neurobiolaging.2021.04.019) showed microvascular degeneration occurring before plaque onset in 3xTg mice. In the present work, it would be interesting to compare the vascular changes with expected plaque load (obtained from other studies or even from the literature). Comparison to that work, showing hippocampal and cortical subregion-dependent changes to microvessels with age in 3xTg AD mice, as well as tortuosity changes, is warranted.

We have added Supplementary Text 11 to discuss this point in detail.

Discussion, line 287:

Finally, the chronological graph showed that most CMDs became apparent between 12 and 25 WOA and preceded the cognitive decline observed from the same animals (Fig. 5c). Some of these CMDs slightly precede even extracellular A β deposits in the 3xTg model (see Supplementary Text 11 for details).

Supplementary information, line 596, new section:

Supplementary Text 11. Comparison to A β pathology in AD.

Various cerebral microvascular degenerations (CMDs) observed in this study became apparent between 12 and 25 WOA (Fig. 5c), preceding extracellular A β deposits in 3xTg model mice. In this specific model, extracellular A β deposits first become apparent in the frontal cortex at 6 months (26 WOA) and then evident in other cortical regions and in the hippocampus by 12 months (52 WOA)¹⁸, and cortical A β plaques were first detected at 12 months of age (52 WOA)⁸. This temporal relationship between microvascular and A β pathologies in the 3xTg model agreed with findings from a recent study using the corrosion cast method⁴⁵. It is difficult to directly compare results between this terminal study and our study, because the terminal study compared absolute values between AD and WT mice, age by age, while our longitudinal study compared the slope of relative changes (RCA) between AD and WT considering all ages of measurement. Nevertheless, we found some consistency. In the somatosensory cortex, the area we investigated, the terminal study found the total length of capillary vessels (5-10 μ m in diameter) was higher at 3 months of age, similar at 6 months, and lower at 12 months in 3xTg compared to WT mice, which is consistent with our result of decreasing capillary length in 3xTg mice (Fig. 3d). The terminal study also found no difference in vessel segment and vessel junction numbers between 3xTg and WT mice across the ages of 3-24 months. This result is agreeable with our result of insignificant differences in capillary branching order, number density, and length density (Table 1). Finally, the terminal study observed more tortuous vessels in 3xTg mice, although it did not conduct quantitative analysis of tortuosity, and we also found an increasing capillary tortuosity with age in 3xTg mice (Table 1).

-The number and diameter of pial collateral “arterioles” decrease with aging is associated with reduced eNOS and increased oxidative stress. Mouse models of AD promote rarefaction of pial collaterals, suggesting inflammation-induced accelerated aging of collateral wall cells (e.g. Zhang H., *Angiogenesis* 2019; 22(2):263-279, doi: 10.1007/s10456-018-9655-0). Despite no difference in pial vessel diameter observed here between WT and 3xTg mice (Fig. 1), has there a difference in pial vessel number been detected? Information about changes in vessel density over the experimental period should be provided.

We appreciate the reviewer's suggestion of investigating pial collateral arteriole density in the context of AD. Although we did not measure the pial collateral arteriole density from OCTA images, we have measured a closely related property, penetrating vessel density from Doppler OCT images. We have added this result to the main text and Supplementary Fig. 3.

Results, line 115, newly added:

Finally, the penetrating vessel density in vessel number per unit area decreased in AD (-6% per month; 95% CI, -11 to -1), but the slopes were not significantly different between AD and WT (p=0.11).

Supplementary information, line 28, updated figure:

Supplementary Figure 3. All properties which either had statistically no difference in the rates of change between AD and WT or had statistically different rates but the fractional changes did not become statistically different between AD and WT until the end of experiment (35 WOA), except for the pial vessel diameter which was shown in Fig. 1.

Minor comments:

-Introduction, line 44: should be "... OCT can rapidly produce microangiograms²¹, ... capillary vessels²³, all in a label-free manner."

We have modified the sentence accordingly.

-Results, line 88: Please specify whether 107 pial vessels were assessed for the different mice measured longitudinally.

We have clarified it.

Results, line 89:

This allowed us to longitudinally track the same set of 107 pial vessels of 13 mice over seven months (gray lines in Fig. 1d).

-Results, line 101: Please specify whether 151 penetrating vessels were assessed for the different mice measured longitudinally.

We have clarified it.

Results, line 102:

This enabled us to longitudinally track the same set of 151 penetrating vessels of 13 mice for seven months (gray lines in Figs. 2c and 2f).

-Results: please specify the meaning of the abbreviation WOA (weeks of age) the first time it appears in the text.

We have specified it.

Results, line 108:

In turn, their fractional changes became significantly different between AD and WT at 18 weeks of age (WOA) ($p < 0.05$, Fig. 2d), which is termed as the age of significance (AOS) hereafter.

-Line 226: should be "... awake imaging as being shown in our follow-up studies ...".

We have modified the sentence accordingly.

-Methods, Animal preparation, lines 365-368: the exact age of the mice at window placement should be provided.

We have clarified it.

Methods, line 425:

Anesthesia and Perioperative Monitoring. At 10 WOA, mice underwent induction of anesthesia with 3% isoflurane in oxygen and were then maintained at 2% isoflurane until recovery in 100% oxygen.

-Line 418: should be "... 147,000 A-scans/s."

We have modified the sentence accordingly.

Reviewer 2

This study provided some interesting advance in age-related alterations in the morphology, topology, and function of the cortical vasculature across all scales, which may be used to monitor the degenerative process of cerebral microvascular disease, especially, in Alzheimer's disease. However, there are some major problems in the reliability of technology, the theory proposed by the authors and statistical methods. This manuscript should be rejected or major revised before publication.

We thank the reviewer for reviewing our manuscript and providing insightful suggestions to improve the manuscript. We have carefully addressed every comment as follows. We have conducted additional experiments to address important comments in a data-backed manner.

We believe that this study has opened and demonstrated a novel methodological possibility for preclinical study of several age-related neurodegenerative diseases. The ability to track cerebral microvascular changes allows for long-term investigations across an unprecedented broad range of vascular properties, including morphological, topological, and functional aspects, spanning all scales of vasculature from pial vessels to capillaries. We hope the reviewer will find our revised manuscript satisfactory for publication.

The problems are listed for reference:

1. Technical reliability

The experimental design had some shortcomings. For example, although the long-term longitudinal observation of vascular degeneration had been achieved by OCT, the inter-day and intra-day variations in this method should be tested in each mouse and provided in this study. Our comments mainly ensued from that these parameters of angiography are most likely affected by eating and drinking, room temperature or stress. Notably, the data in Figure 2c, 2f showed too much dispersion in WT group. Such discrete data are difficult or impossible to provide recognizable differences in actual imaging for diagnosing disease.

As the reviewer pointed out, and as we had discussed in Supplementary Text 3 of the original manuscript, the measurement result exhibited large vertical dispersion (gray lines representing individual vessels in Figs. 2c and 2f). While we acknowledge that this dispersion could pose challenges in applying the findings directly to human AD diagnosis, we want to emphasize that the main focus of this study is to demonstrate the novel technical capabilities as a preclinical research tool, as we had clarified in Supplementary Text 4 of the original manuscript. Therefore, first, we have further clarified this scope of the presented work as follows.

Discussion, line 252:

In our results with the 3xTg model (see Supplementary Text 4 for use of the specific model within the scope of this study), the earliest cerebral microvascular degeneration (CMD) was detected in the diameter of penetrating vessels (thinner in AD mice), followed by degeneration in their blood flow (lower in AD).

Supplementary information, line 280, new paragraph:

First, it is important to note that the aim of this study is to demonstrate the technical capabilities of our approach as a preclinical research methodology, rather than directly addressing biological questions or developing a diagnosis technique for human AD. While Supplementary Text 3 briefly discussed the methodological parallelism and noted potential clinical applicability of our method, the validation of this applicability will require a set of additional research, such as cortical imaging in clinically more relevant models that account for a larger population of human AD (e.g., late-onset AD with ApoE4), longitudinal retinal imaging in such models, and a small-scale proof-of-concept human study, prior to conducting a large-scale clinical trial.

Within the clarified scope, we have conducted an additional experiment to directly address the concern on the **intra-day variation** first. Intra-day variation can be affected by many factors, including technical stability in imaging (both imaging hardware and analysis software) and physiological stability of animals (e.g., fluctuating stress level during imaging). In the additional experiment, we have tested how robust our approach is; specifically, whether vascular measures with the approach robustly comes back to the baseline when an animal undergoes vasodilation and then recovers from it. To reduce a false-negative error in this test, we have also measured the sensitivity of the approach, i.e., how much small changes in vascular measures the approach can detect. In this response, we refer to “our approach” by the combination of longitudinal OCT imaging and its analysis with linear mixed-effect (LME) models, the same approach we had used in the main study. The additional experiment found that our approach is sensitive enough to detect changes in vessel diameter down to 1.4 μm (7% change) while being robust enough to produce statistically *insignificant* measures between the pre-vasodilation and post-recovery stages ($p>0.05$). We have added Supplementary Text 10 and Supplementary Fig. 12 to describe details of this additional experiment and related discussion. Please see these newly added supplementary information (copied below) to see how the LME-analyzed longitudinal measurement can be robust against large vertical dispersion of individual traces.

Discussion, line 247:

Supplementary Text 3 provides further discussion of other challenges and opportunities related to the presented methods. Supplementary Text 10 discusses the sensitivity of our approach to detecting changes in vascular diameter and blood flow.

Supplementary information, line 532, new section:

Supplementary Text 10. Sensitivity of OCT measurements of changes in vascular diameter and blood flow.

Although the statistical results presented in the main text support that our approach was sensitive enough to reveal subtle differences in those vascular changes with age between AD and WT mice, we conducted a separate experiment to directly quantify the sensitivity of our approach of detecting changes in the average diameter and blood flow of pial vessels and penetrating arterioles.

Methods: We used isoflurane as a vasodilator and repeated OCT imaging while vessels went through dynamic changes via the dilation and recovery. In detail, seven days after we installed a chronic cranial window on a mouse as described in the main text, we put the mouse on an air-floating platform (Mobile Homecage, Neurotar) for a head-fixed, freely-walking awake imaging condition, while the mouse nose is connected to the isoflurane vaporizer. We acquired nine sets of OCT angiogram and Doppler OCT images, following the exact protocols described in the main text, at the following time points:

1. Awake as a baseline
2. 5 minutes after turning ON isoflurane supply
3. 25 minutes after turning ON isoflurane supply
4. Right after turning OFF isoflurane supply
5. 7 minutes after turning OFF isoflurane supply
6. 15 minutes after turning OFF isoflurane supply
7. 30 minutes after turning OFF isoflurane supply
8. 45 minutes after turning OFF isoflurane supply
9. 60 minutes after turning OFF isoflurane supply

This longitudinal dataset underwent the analysis process as described in the main text, in order to identify same vessels across time points and measure the pial vessel diameter, arteriolar diameter and flow, and venular diameter and flow, for each vessel and each time point. We repeated this data acquisition and analysis in four young WT mice. For statistical analysis of a change between two states, we used linear mixed-effect (LME) analysis with the nested random effects of individual vessels in animals. This nested LME was identical to the LME analysis used in the main text.

Results: As expected, the pial vessel diameter increased with isoflurane and returned to the baseline after recovery (Supplementary Fig. 12a). Our approach was sensitive enough to detect a detectable 7% change in the pial vessel diameter between the first awake state and the state 5 minutes after turning on the isoflurane supply. The average pial vessel diameter in the awake state was 19.7 μm , and the increased diameter in the second state was 1.4 μm ($p=0.026$, $n=59$ vessels from four animals; Supplementary Fig. 12b). **Our approach was also robust enough to produce statistically insignificant results between the first and the final states as physiologically expected ($p=0.932$, Supplementary Fig. 12c) while still detecting the 1.4- μm change.**

Similarly, the arteriolar diameter and flow increased with isoflurane and returned to the baseline after recovery, while the venular diameter and flow did not (Supplementary Fig. 12d). We found subtle fluctuations during the recovery phase, the degree of which our approach was able to detect. The average arteriolar diameter in the first of the final two

states (45 minutes after turning off isoflurane supply) was 27.5 μm , and the increased diameter in the final state was 1.6 μm , a 6% change ($p=0.022$, $n=7$ arterioles from four animals; Supplementary Fig. 12e, top). The average arteriolar flow changed by 0.098 $\mu\text{L}/\text{min}$ from 0.409 $\mu\text{L}/\text{min}$, a 24% change ($p=0.052$, Supplementary Fig. 12e, bottom).

Our approach was also robust in producing statistically insignificant results between the first and the final states as biologically expected ($p=0.942$ and 0.436 for diameter and flow, respectively; Supplementary Fig. 12f).

Discussion: The supplementary experiment presented here provides a quantitative measure of the sensitivity of our approach for detecting changes in vascular diameter and blood flow. We found the sensitivity of detecting changes in pial vessel diameter, arteriolar diameter, and arteriolar blood flow to be 1.4 μm (7% change), 1.6 μm (6% change), and 0.098 $\mu\text{L}/\text{min}$ (24% change), respectively. It is important to note that these sensitivity values serve as a conservative estimate of the actual sensitivity of the measurements described in the main text. This is because the supplementary experiment used fewer animals (four, against 6-7 per group of the main study) and compared only two time points, whereas the main experiment employed seven time points to determine the slope (rate of change with age, RCA). It is likely that the RCA measurement described in the main text has higher statistical power and thereby higher sensitivity.

Supplementary information, line 132, new figure:

Supplementary Figure 12. OCT-measured changes in vascular diameter and flow during vasodilation and recovery. **a)** Tracked vessel diameter over 59 same pial vessels in four animals, with the blue line indicating the simple average of the diameters for each time point. **b)** Pial vessel diameter changes between the awake state and the state after 5 minutes of turning on isoflurane supply. The selected time points produced a p value close to 0.05. **c)** Pial vessel diameter changes between the first state (awake) and the final state (after 60 minutes of turning off isoflurane supply). **d)** Tracked vessel diameter and flow over seven same arterioles and 13 same venules in four animals. **e)** Arteriolar diameter and flow changes between the state before the final state (45 minutes after turning off isoflurane supply) and the final state, with the selected time points producing p values close to 0.05. **f)** Arteriolar diameter and flow changes between the first and final states.

Regarding the **inter-day variation**, it can be affected by many factors including eating and drinking as the reviewer pointed out, and it will contribute to the vertical dispersion of individual vessel measurements in the main study (gray lines in Figs. 1 and 2). The question is whether or not our approach is robust enough to support the conclusion against such random-wise inter-day variations. First, the additional experiment above shows that our approach is robust and sensitive against a large vertical dispersion (gray lines in Supplementary Fig. 12), and this dispersion is similar to that of the inter-day variations we observed in the main study (gray lines in Figs. 1 and 2). Therefore, it is logical to infer that our approach is also robust and sensitive against inter-day variations as well. Second, the statistical significance presented in the main study (Table 1) supports that the approach was sensitive and robust enough to detect the reported subtle changes and to distinguish between significantly and insignificantly varying vascular properties, despite the observed dispersion.

In summary, this technical reliability against dispersion (higher than expected from the visual dispersion) could be attributed to three factors:

- a. The longitudinal nature of our data: If we measured vessel diameters from different animals of different ages and compared them, the concerned vertical dispersion would directly determine the statistical power. This approach of analysis is traditionally used in many *terminal* studies, and we understand that the visual dispersion can raise the concern. However, this direct relation between dispersion and statistical power is not always true for *longitudinal* data. For example, the LME analysis used in the present study properly handles subject-specific effects, which can significantly mitigate the impact of vertical dispersion, as we had discussed in Supplementary Text 3 of the original manuscript. As an illustrative example, a video from MATLAB¹ clearly shows why one should consider such random effects in analyzing longitudinal data with LME and how it is strongly robust against vertical dispersion.
- b. Sensitivity of the approach: Noise in the vessel diameter/flow measurement can arise by many factors, including the physiological factors such as eating and drinking, room temperature, or stress, as noted by the reviewer. Despite this noise with various origins, the sensitivity of our approach has shown high enough to detect subtle differences in the vessel diameter/flow.
- c. Robustness of the approach: Beside those physiological factors, some technical factors can generate additional noise, even in the measurements made from the same animal in the same day, as we had briefly discussed in Supplementary Text 3 of the original manuscript. However, the additional experiment has also shown that our approach is robust against such intra-day random variations.

2. Theoretical reliability

The conclusion of temporal relationships between microvascular degenerations and cognitive impairment needs more sufficient evidences.

¹ <https://www.youtube.com/watch?v=-XVVjwSqbZo>

Although the authors found some differences in microvascular parameters between wild-type and AD model mice, this manuscript did not provide the convincing evidence, but only a novel object recognition experiment with no valid correlation analysis. The experiments of Morris water maze and fear conditioning test should be provided, and the relationship between the parameters of microvascular degeneration and the degree of cognitive decline should be investigated.

As a quick clarification, the temporal relationship between two events does not necessarily mean a correlation between them. By the term “temporal relationship” in the original manuscript, we referred to “whether and how long a type of alteration becomes apparent earlier than another”, where the age of being apparent was defined by the age of significance (AOS) in the manuscript.

We appreciate the reviewer's suggestion for investigating the correlation between the parameters of cerebral microvascular degeneration (CMD) and the degree of cognitive decline. While the purpose of our study was to demonstrate a novel technical capability, we agree that exploring the CMD-cognition correlation can be of additional interest, as long as we clarify one must be careful in translating the finding to human AD. This required careful consideration had been already discussed in the original manuscript in its Supplementary Text 4.

In response to this suggestion, we have conducted an additional analysis to measure the correlation between the NOL score and each of the 25 vascular properties. The results show that a subset of these properties is significantly correlated with the NOL score. We have replaced the original Fig. 5d with this new analysis result and moved the original Fig. 5d to Supplementary Fig. 8. The original Fig. 5d presented the correlation among CMDs, not between CMD and NOL. We have updated the related parts of the Results, Discussion, and Methods sections accordingly. We also have added a paragraph to Supplementary Text 4 to further clarify that the newly-added correlation result does not necessarily mean the causality.

Results, line 199, new paragraph:

To reveal the correlation between the simultaneously observed vascular alterations and cognitive impairment, we calculated the correlation coefficient with age lags of 0, 4, 8 and 12 weeks in the AD group (Methods). Our analysis found that 19 vascular properties were significantly correlated with the NOL discrimination index (Fig. 5d). Some correlations were expected, such as the positive correlations between the decrease in NOL test score and the decreases in arteriolar and venular diameter and flow. Other correlations were unexpected; for example, the betweenness was strongly negatively correlated with the NOL score. This betweenness result was interesting when considering that it exhibited the youngest AOS among capillary vessel properties (Fig. 5c) and formed the greatest number of inter-correlations with other vascular alterations (Supplementary Fig. 8, see Discussion for interpretation).

Discussion, line 218:

Among other results presented here, the chronological and correlation graphs between cognitive impairment and vascular alterations (Figs. 5c and 5d) are a representative example of what type of information can be obtained by the presented methods. Such information may provide insight into how cognitive impairment and different types of vascular alterations develop along with or independently of each other.

Discussion, line 287, new paragraph:

Finally, the chronological graph showed that most CMDs became apparent between 12 and 25 WOA and preceded the cognitive decline observed from the same animals (Fig. 5c). Some of these CMDs slightly precede even extracellular A β deposits in the 3xTg model (see Supplementary Text 11 for details). The correlation graph (Fig. 5d) showed that 19 vascular properties were significantly correlated with the cognitive NOL test score. While some of these confirmed expected correlations, such as those of arteriolar and venular diameter and flow, others revealed new findings. Of particular interest is the betweenness property, which displayed a strong negative correlation with the NOL test score, meaning that higher betweenness was associated with later cognitive decline. It was the earliest capillary vessel property to show significant differences between AD and WT mice (Fig. 5c) and was the most inter-correlated with other vascular alterations in both AD and WT, but with different inter-correlation patterns (Supplementary Fig. 8). These interesting findings suggest that the betweenness of the capillary network is an important component of CMD in neurodegenerative diseases. This spotlight on the relatively novel topological property, now longitudinally measurable in vivo using the presented methods, encourages further studies on how other vascular and non-vascular factors interact with capillary network betweenness and how this interaction contributes to the pathology of vascular and other related systems in aging research.

Figures, line 387, updated figure:

d) Significant correlations between the NOL discrimination index and the vascular alterations ($p < 0.05$, see Methods for details). Pairs with positive and negative correlations are shown in red and blue respectively. More opaque lines indicate higher maximum correlation, and thinner lines mean that the maximum correlation appeared at longer age lags. A correlation with an age lag means that one alteration was correlated to the other one with a certain time delay. Two examples of significant correlations are presented on the right, where each circle represents a measurement from a single animal at a single age point, and the orange line and shade depict the fitted line and its 95% CI.

Methods, line 618, new paragraph:

To study the correlation between cognitive decline and the vascular features studied, we fitted LME models to determine the relationship between the fractional change of each vascular feature and NOL discrimination index. Two such examples are demonstrated on

the right of Fig. 5d. We fitted these models while also introducing time-lags of 0, 4, 8 and 12 months. The features which exhibited a significant slope ($p < 0.05$) were considered to be correlated with cognitive function, with the sign of the slope indicating positive or negative correlation.

Finally, as for employing multiple cognitive tests, we agree that additional behavioral tests would have further enhanced rigor of our measurement of cognitive impairment. When we had designed this study, however, we decided to use the NOL test alone because of three reasons.

- a. High risk on the 7-month longitudinal experiment: At the design stage, our animal behavioral facility did not recommend using the Morris water maze due to potential water damage to the cranial window headpost. Since no study had validated how long OCT imaging via a cranial window could generate images with acceptable quality, we decided not to take the risk. We were also aware of the study reporting that the Morris water maze test was less sensitive in detecting age-related cognitive decline in AD model mice².
- b. Animal welfare: Fear conditioning could have been a potential addition, but at the design stage, we were unable to find a study that conducted fear conditioning tests repeatedly on the same animal over a long period of several months. Since most of the fear conditioning tests have a period of days, addition of a fear conditioning test to our longitudinal study design would have required us to repeat a set of conditioning and testing phases every month, which could have caused an issue in animal welfare. Furthermore, there is a risk of applying electric shocks to an animal with a metal head post attached.
- c. Well characterized cognitive decline in 3xTg model: After two decades from their generation, 3xTg-AD mice are still one of the most widely used transgenic models of AD, and their cognitive decline has been relatively well characterized in the literature. For example, Belfiore et al. found spatial learning and memory deficits at 6 months of age (26 WOA)³, which is highly consistent with our result from the NOL test (27 WOA, Fig. 5b).

We have added this discussion to Supplementary Text 3.

Supplementary information, line 211:

A related challenge in a longitudinal imaging experiment with the cranial window is a limitation in employing behavioral tests to simultaneously perform on the same set of animals. In this study, we employed the NOL test alone for several reasons. First, the Morris water maze test, widely used in this context, could pose a risk of failure due to potential water damage to the cranial window headpost. Second, fear conditioning could be a potential addition to the NOL test, but repeatedly subjecting the same animal to conditioning and testing phases for several months could raise animal welfare concerns.

² Kishimoto Y, Higashihara E, Fukuta A, Nagao A, Kirino Y. Early impairment in a water-finding test in a longitudinal study of the Tg2576 mouse model of Alzheimer's disease. *Brain Res.* 2013 Jan 23;1491:117-26. doi: 10.1016/j.brainres.2012.10.066. Epub 2012 Nov 7. PMID: 23142630.

³ Belfiore R, Rodin A, Ferreira E, Velazquez R, Branca C, Caccamo A, Oddo S. Temporal and regional progression of Alzheimer's disease-like pathology in 3xTg-AD mice. *Aging Cell.* 2019 Feb;18(1):e12873. doi: 10.1111/acel.12873. Epub 2018 Nov 28. PMID: 30488653; PMCID: PMC6351836.

Additionally, applying electric shocks to an animal with a metal head post attached could be a potential issue. Finally, the 3xTg model used in this study is one of the most widely used transgenic models of AD, and its cognitive decline has been well characterized in the literature. For example, Belfiore et al. found spatial learning and memory deficits at 6 months of age (26 WOA)⁸, which is highly consistent with our result from the NOL test (27 WOA, Fig. 5b).

The minor problems:

1. Please provide details of the method of surgery for the chronic cranial window, including the specific location of the cranial window. How to make sure that the cranial window does not heal during the long experiment, thereby ensuring good imaging quality, should be explained or discussed it.

The original manuscript already described details of the surgical method in its Method section as follows. We have added further details to the part.

Methods, line 439:

Craniotomy. After induction of anesthesia, the scalp overlying the parietal bones was shaved and then sterilized using alcohol and iodine scrub. A 1-cm midline incision was made on the scalp, and the skin on both sides of the midline was retracted. The pericranial tissue was stripped using needle-tip forceps. A proprietary metal frame, used to hold the head and prevent motion during microscopy (Supplementary Fig. 11), was affixed to the skull with dental cement (Parkell Inc., Long Island, NY, USA). A 3-mm bone flap overlying the left parietal cortex was thinned with a dental burr until transparent (100- μ m thickness). In detail, the location of the cranial window was identically selected in all mice, such that the medial border of the window was 1mm lateral to the sagittal suture and the posterior border was 1-mm anterior to the lambdoid suture on the right side, allowing for visualization for visual, somatosensory, and motor cortex. The thinned bone flap was removed with fine-tip forceps while keeping the dura intact and the skull defect filled with normal saline until hemostasis was achieved. Our circular glass cranial window detailed below was then placed into the defect. The window was then fixed in place and the annular skull defect surrounding the window was sealed using dental cement. The cement was given time to fully cure, and the mouse was then recovered in an oxygen chamber. Mice were housed in individual cages postoperatively to protect the integrity of the metal frame.

The base of the window consisted of two 3-mm glass cover slips and the apex consisted of one 5-mm circular glass slip; these were epoxied together with a transparent UV-cured resin. The thickness of the 3mm-wide base of the window was 0.24-mm (two stacked cover slips of 0.12-mm thickness each), closely approximating the thickness of the mouse parietal bone (~0.2 mm)³⁹. The 5-mm cover slip serving as the apex of our window acted

as an additional safety measure against compression of the cortex, as it was wider than our 3-mm craniotomy and thus any pressure against the window prior to cementing was applied to the bone surrounding the craniotomy and not the brain tissue itself; it also ensured that the base of the window sat flush just above the cortex without compressing it. Once the window was cemented in place, this minimized any dural scarring or scar tissue ingrowth underneath the window for the duration of our experiment.

Despite this effort to minimize dural scarring or scar tissue growth, three out of 19 surviving animals exhibited the imaging quality that degraded significantly with time, as clarified in the original manuscript. Scar tissue formation underneath the window prevented a sufficient number of vessels from being visible in OCT images; these animals were excluded from analysis, as clarified in the original manuscript.

To further address this comment, especially how to make sure that the cranial window does not produce abnormal immunoreactivity during the long experiment, we have acquired additional data with post-mortem immunofluorescence imaging of the brains of the animals that had undergone the longitudinal experiment. We fortunately had collected and fixed the brains of the animals after the longitudinal experiment. Here, we have used these saved brains for immunofluorescence imaging to investigate the possible side effects of the window. We have added Supplementary Text 8 and Supplementary Fig. 9 to describe this additional experiment and its result.

Supplementary information, line 424, new section:

Supplementary Text 8. Potential impact of long-term craniotomy and imaging on animal physiology.

As described in the Methods section, we started with 20 animals, but one animal did not survive until the end of our seven-month longitudinal experiment. Three animals were euthanized in the middle of the study due to mechanical damage to their headposts, and three animals were excluded due to degrading image quality. The degrading image quality (3 out of 20) was first noticeable as blurry capillaries in microangiograms, likely due to the formation of a thin layer of scar tissue below the bottom surface of the glass and the top surface of the cortex. The mechanical damage (3 out of 20) occurred while the animals were moving in their cages. The spontaneous death rate (1 out of 20, 5%) was comparable to the known survival rate at the corresponding age⁴⁰. This consistent survival rate suggests that chronic cranial windows might not have a severe long-term influence on the physiology of animals.

Several studies have investigated the effect of open-skull craniotomy on cortical physiology, particularly in terms of immunoreactivity. The literature suggests that the craniotomy may lead to higher immunoreactivity within 3-4 weeks after installation, but the immunoreactivity returns to normal after that time. For example, Xu et al. reported higher immunoreactivity in the window-installed cortex than the other side within 20 days of installation, but no difference at 30 days⁴¹. Holtmaat et al. also reported that astrocytes, not microglia, became more immunopositive in the window-installed cortex at 2 weeks but returned to normal at 4 weeks of window installation⁴². Similarly, Goldey et

al. reported no difference between the ipsilateral and contralateral cortices of window installation at about 10 days⁴³. Heo et al. found that a PDMS window installation led to a high microglia cell density, not astrocytes, at 1 week but the density returned to normal at 3 weeks⁴⁴.

Although the spontaneous death rate and existing literature suggest that long-term craniotomy is unlikely to have adverse effects on animal physiology, there has been no direct investigation into this possibility. To address this, we conducted post-mortem immunofluorescence imaging of the brains of animals that had undergone our longitudinal experiment. Using the published method of Heo et al.⁴⁴, we fixed, sectioned, and stained the brains with GFAP and DAPI (Supplementary Fig. 9a) or IBA-1 and DAPI (Supplementary Fig. 9d), which respectively visualize astrocyte and microglia immunoreactivity. These two stains are widely used in the literature to investigate the effect of craniotomy on cortical physiology⁴¹⁻⁴⁴. We measured astrocyte immunoreactivity by calculating the ratio of GFAP to DAPI in terms of cell number or pixel area, and found no statistically significant differences between the ipsilateral and contralateral cortices of the window installation in both ratios (Supplementary Fig. 9c). We performed similar analysis on the DAPI and IBA-1 stained images and found no statistically significant differences in microglia immunoreactivity (Supplementary Fig. 9f).

Supplementary information, line 92, new figure:

Supplementary Figure 9. Post-mortem immunofluorescence result. **a)** An example of GFAP and DAPI stain images. The yellow polygons indicate the regions of interest (ROIs) used for quantitative analysis in (c). Scale bar, 1 mm. **b)** Magnified images of the white rectangular areas in (a). Scale bar, 0.1 mm. **c)** 17 slices were analyzed (12 slices from 3 AD mice and 5 slices from 2 WT mice). All GFAP images underwent an identical image segmentation process, which used adaptive thresholding and filtered foreground objects by the area (equivalent to circles of 10-30 μm in diameter). DAPI images underwent a circle detection processing. The differences were normally distributed, and we used paired t-tests to analyze them. We excluded one slice as an outlier in the area ratio analysis. The effect of AD on this conclusion of insignificant difference was not statistically significant ($p=0.193$, number ratio; 0.267 , area ratio; linear mixed-effect [LME] analysis). **d)** An example of IBA-1 and DAPI stain images. The yellow polygons indicate the ROIs used for quantitative analysis in (f). Scale bar, 1 mm. **e)** Magnified images of the white rectangular areas in (d). The IBA-1 channel images underwent contrast adjustment to suppress background fluorescence. **f)** 13 slices were analyzed (9 slices from 2 AD and 4 slices from 1 WT mice). All IBA-1 images underwent the identical image segmentation as that for GFAP images. DAPI images also underwent the identical circle detection processing. The differences were normally distributed, and we

used paired t-tests to analyze them. The effect of AD on this conclusion of insignificant difference was not statistically significant ($p=0.093$, number ratio; 0.054 , area ratio; LME analysis).

2. In Figure 1b, an example of selecting the same vessel across time points needs to be provided. Please explain: How to ensure that the same position of the same blood vessel was selected at different time points? Whether the diameter of blood vessels changes during the course of running or not? Why only a single cross-sectional diameter was used to represent the thickness of the blood vessel?

The original Fig. 1b already included an example of selecting the same vessels across time points (the color lines in circles), but the figure caption lacked detailed explanation. We have improved the figure caption as follows.

Figures, line 325:

b) An example of selecting the same vessels across time points. The color lines in circles indicate the selected vessels. To ensure selecting the same vessel across time points, we considered its relative position within the vascular branches, as visually shown in this example. Image registration, as shown in (a), facilitated this visual inspection and vessel selection. Each of the color lines is drawn along the automatically detected line orthogonal to the orientation of the selected vessel, along which the cross-sectional intensity profile was extracted for diameter measurement.

In response to the question regarding whether the diameter of blood vessels changes during the course, we have interpreted it as whether the vessel diameter physiologically varies during the period of image acquisition. We agree that vessel diameter naturally fluctuates according to heart beat, which could affect measurements if a single snapshot is used. For this reason, we acquired 10 volumes and used the averaged image to minimize the effect of periodic fluctuations. We have clarified this in the Methods section.

Methods, 499:

During each imaging session, we acquired angiograms of large pial vessels using a 5X objective lens, with a field of view (FOV) of $1024 \times 1024 \times 512$ (x,y,z) corresponding to an imaging volume of about $3 \times 3 \times 1.8$ mm. Ten angiogram volumes were acquired, motion-corrected⁴¹, and then volume-averaged prior to diameter measurements, in order to minimize the effect of physiological cardiac fluctuations.

Regarding the third question on why only a single cross-sectional diameter was used to represent the thickness of the blood vessel, we want to clarify that we did not use a single cross-sectional profile when measuring the diameter of a vessel. To minimize variations in vessel thickness along the centerline, our software automatically identified the closest point on the centerline of the vessel, extracted 10 cross-sectional profiles from 10 adjacent pixels along the centerline, and averaged them before fitting the

averaged profile to a Gaussian function. We presented this final step of fitting in the original Fig. 1c. We have added these intermediate steps to the figure caption.

Figures, line 331:

c) An example of the cross-sectional profile and its fitting to a Gaussian function to measure the diameter as the full width at half maximum. To make the diameter measurement robust against slight fluctuations in vessel thickness along the vessel centerline, ten adjacent cross-sections were extracted around the selected location and then averaged along the vessel centerline, prior to the Gaussian fitting.

3. Please provide the acquisition time of all mice at each time point.

We have added this information to the Methods section.

Methods, line 499, newly added:

Each OCT imaging session took about an hour per mouse; 10 minutes for angiogram, 5 minutes for Doppler, 10 minutes for microangiogram, and 30 minutes for RBC data. We held imaging sessions routinely during the daytime.

4. The results of the trans-genetic mice should be normalized to that of WT mice at each time point, so as to eliminate the influence of instruments and measurement conditions.

We appreciate the suggestion for data normalization, but we respectfully disagree that it is the most appropriate approach for our longitudinal dataset. While normalization may be effective in terminal experiments, it may not be suitable for analyzing longitudinal data where subject-specific random effects can significantly influence the normalization. For instance, some animals may inherently have higher or lower baseline values than others within both the WT and AD groups, resulting in large vertical variations as discussed in our response to the reviewer's first comment above. Normalization to the WT group mean at each time point could lead to over- or under-normalization of some AD animal values, compromising the validity of our results. Instead, we used linear mixed effects models (LME) that allow for subject-specific random effects and can handle the inherent differences in baselines across animals. As shown above, this approach is strongly robust against vertical variations. Finally, regarding the specific influence of instruments and measurement conditions concerned by the reviewer, our analysis compared the slope of the LME fit (the rate of change with age, RCA) between AD and WT, neither comparing the absolute slope for each group nor comparing measurement values at each time point. Thus, our analysis is relatively free from the concerned influence.

5. The manuscript mainly studied image registration and deep learning, etc. Please supplement the physiological significance of the research.

As the reviewer correctly pointed out, the major aim of this study is to validate the claimed technical capability rather than to make novel physiological findings. Following this suggestion, we have added physiological interpretation and significance to several relevant locations throughout the manuscript.

Discussion, line 261, newly added:

Although the primary focus of our study is technical demonstration rather than biological discovery (Supplementary Text 4), the observed cortical hypoperfusion in AD is consistent with findings from perfusion magnetic resonance imaging and transcranial Doppler ultrasound studies, which have reported hypoperfusion in various cortical regions in early human AD (see Refs. 33 and 34 for reviews).

Supplementary information, line 368, new paragraph:

Our results showing a decrease in mean capillary length with aging in AD (Fig. 3d) are consistent with a previous study that found capillary length to be shorter in AD mice than in wild-type (WT) mice at 18-31 WOA⁷, although that study did not longitudinally track the length and used a different AD mouse model. Another study found that capillary branch numbers were lower in 3xTg mice than WT mice at 20 months of age, but no difference was observed at 7 and 14 months²⁶. This finding is consistent with our result of shorter capillary lengths in 3xTg mice, but our measurement revealed the difference much earlier (Fig. 3d). There are several possible reasons for this discrepancy, including that (i) the statistical analysis in the previous study did not consider the clustered nature of the data, leading to lower sensitivity to intra-group differences; (ii) our method tracks the trend of the capillary property varying with age, whereas their method relies on a snapshot at a specific age; and/or (iii) our method analyzes 3D networks, while their method analyzes only 2D cross-sections of the network.

Supplementary information, line 396, new paragraph:

The combined results of arteriolar diameter/flow and capillary flow suggest another possible consequence of decreased arteriolar diameter and flow, namely an increase in the presence of hypoxic micro-pockets in the cortex. The observed decreases in both arteriolar diameter and flow would lead to arteriolar and near-arteriolar tissue oxygen pressure (pO₂)³⁴. This impaired arteriolar oxygen delivery would not immediately lead to an impairment in capillary-bed tissue pO₂ when the compensatory mechanism by enhanced capillary flow works as hypothesized above from our observed capillary RBC flux increases. However, when the compensatory mechanism no longer works at later ages, the capillary-bed tissue pO₂ would become lower and spatially more heterogeneous, as observed between 60 and 100 WOA in WT mice³⁴. This impaired capillary oxygen supply increases the presence of hypoxic micro-pockets, which may explain the observation of tinier microinfarcts in aging brains, particularly those with mild cognitive decline or AD^{35,36}. Therefore, the decrease in arteriolar diameter and flow observed in both WT and 3xTg mice with age may have important implications for the development and progression of neurodegenerative diseases.

Discussion, line 284:

The presented results also showed degenerations in capillary blood flow (see Supplementary Text 7 for interpretation and potential relation to hypertension).

Supplementary information, line 411, new paragraph:

The decrease in penetrating arteriole and venule diameters (Table 1) further suggests that the blood pressure might have increased with age in the 3xTg mice we used. Unfortunately, we did not longitudinally measure blood pressure in our study. Hypertension has been linked to AD, and animal studies suggest that hypertension can lead to amyloid plaques, neuroinflammation, blood-brain barrier dysfunction, and cognitive impairment³⁷. However, human studies on the association between hypertension and AD have been sparse and inconsistent, and the effect of antihypertensive medications on AD seems weak^{38,39}. Investigating interactions among plaques, tangles, cerebrovascular pathology, and dementia may be key to understanding hypertension's role in AD development³⁹. Therefore, our future work includes performing a long-term trace of blood pressure using a noninvasive method like the tail-cuff technique in diverse AD models, which will enable us to compare the long-term blood pressure trace with the traces of cerebral microvascular properties obtained by the presented methods.

Discussion, line 288:

Finally, the chronological graph showed that most CMDs became apparent between 12 and 25 WOA and preceded the cognitive decline observed from the same animals (Fig. 5c). Some of these CMDs slightly precede even extracellular A β deposits in the 3xTg model (see Supplementary Text 11 for details).

Supplementary information, line 596, new section:

Supplementary Text 11. Comparison to A β pathology in AD.

Various cerebral microvascular degenerations (CMDs) observed in this study became apparent between 12 and 25 WOA (Fig. 5c), preceding extracellular A β deposits in 3xTg model mice. In this specific model, extracellular A β deposits first become apparent in the frontal cortex at 6 months (26 WOA) and then evident in other cortical regions and in the hippocampus by 12 months (52 WOA)¹⁸, and cortical A β plaques were first detected at 12 months of age (52 WOA)⁸. This temporal relationship between microvascular and A β pathologies in the 3xTg model agreed with findings from a recent study using the corrosion cast method⁴⁵. It is difficult to directly compare results between this terminal study and our study, because the terminal study compared absolute values between AD and WT mice, age by age, while our longitudinal study compared the slope of relative changes (RCA) between AD and WT considering all ages of measurement. Nevertheless, we found some consistency. In the somatosensory cortex, the area we investigated, the terminal study found the total length of capillary vessels (5-10 μ m in diameter) was higher at 3 months of age, similar at 6 months, and lower at 12 months in 3xTg compared

to WT mice, which is consistent with our result of decreasing capillary length in 3xTg mice (Fig. 3d). The terminal study also found no difference in vessel segment and vessel junction numbers between 3xTg and WT mice across the ages of 3-24 months. This result is agreeable with our result of insignificant differences in capillary branching order, number density, and length density (Table 1). Finally, the terminal study observed more tortuous vessels in 3xTg mice, although it did not conduct quantitative analysis of tortuosity, and we also found an increasing capillary tortuosity with age in 3xTg mice (Table 1).

Discussion, line 287, new paragraph:

Finally, the chronological graph showed that most CMDs became apparent between 12 and 25 WOA and preceded the cognitive decline observed from the same animals (Fig. 5c). Some of these CMDs slightly precede even extracellular A β deposits in the 3xTg model (see Supplementary Text 11 for details). The correlation graph (Fig. 5d) showed that 19 vascular properties were significantly correlated with the cognitive NOL test score. While some of these confirmed expected correlations, such as those of arteriolar and venular diameter and flow, others revealed new findings. Of particular interest is the betweenness property, which displayed a strong negative correlation with the NOL test score, meaning that higher betweenness was associated with later cognitive decline. It was the earliest capillary vessel property to show significant differences between AD and WT mice (Fig. 5c) and was the most inter-correlated with other vascular alterations in both AD and WT, but with different inter-correlation patterns (Supplementary Fig. 8). These interesting findings suggest that the betweenness of the capillary network is an important component of CMD in neurodegenerative diseases. This spotlight on the relatively novel topological property, now longitudinally measurable in vivo using the presented methods, encourages further studies on how other vascular and non-vascular factors interact with capillary network betweenness and how this interaction contributes to the pathology of vascular and other related systems in aging research.

6. Figure 1 displayed an example set of OCT angiograms to demonstrate that the technique is stable enough for longitudinal imaging. It will be more convincing if giving statistical results of quantitative parameters, such as the average intensity of one certain vessel at different time.

We agree that it would be more convincing if we quantitatively tested the stability or robustness of our measurement presented in Fig. 1. This was another reason for us to have conducted the additional experiment described in the response to the reviewer's first comment above on the technical reliability (Supplementary Text 10 and Supplementary Fig. 12). With the additional data, we have tested the stability/robustness of our repeated OCT imaging. This test focused on answering a question of whether the OCT-measured pial vessel diameter is not significantly different between two repeated imaging sessions that are separated by dynamic vasodilation and recovery during 90 minutes, under the assumption that the recovery period of 60 minutes is long enough for the physiology to truly return to the baseline state. We found that our approach is robust enough to produce statistically insignificant

results between the two repeated sessions ($p=0.932$, $n=59$ vessels from four animals; Supplementary Fig. 12c) while it is sensitive enough to detect diameter changes down to $1.4\ \mu\text{m}$ ($p=0.026$, Supplementary Fig. 12b).

Below is the related part of the added Supplementary Text 10.

Supplementary information, line 570:

Our approach was also robust enough to produce statistically insignificant results between the first and the final states as physiologically expected ($p=0.932$, Supplementary Fig. 12c) while still detecting the $1.4\text{-}\mu\text{m}$ change.

Supplementary information, line 581:

Our approach was also robust in producing statistically insignificant results between the first and the final states as biologically expected ($p=0.942$ and 0.436 for diameter and flow, respectively; Supplementary Fig. 12f).

Supplementary information, line 132, new figure:

Supplementary Figure 12. OCT-measured changes in vascular diameter and flow during vasodilation and recovery. **a)** Tracked vessel diameter over 59 same pial vessels in four animals, with the blue line indicating the simple average of the diameters for each time point. **b)** Pial vessel diameter changes between the awake state and the state after 5 minutes of turning on isoflurane supply. The selected time points produced a p value close to 0.05. **c)** Pial vessel diameter changes between the first state (awake) and the final state (after 60 minutes of turning off isoflurane supply). **d)** Tracked vessel diameter and flow over seven same arterioles and 13 same venules in four animals. **e)** Arteriolar diameter and flow changes between the state before the final state (45 minutes after turning off isoflurane supply) and the final state, with the selected time points producing p values close to 0.05. **f)** Arteriolar diameter and flow changes between the first and final states.

7. The left figure in Figure 4a could be displayed in different colors, similar to the right figure, to provide better visual effect.

In Fig. 4a, it is not feasible to use color coding in the left figure like the one used in the right figure. This is because the colors in the right figure represent the uncertainty of the CNN estimation, while the traditional peak-counting method presented in the left figure does not provide the uncertainty of its estimation. The information about uncertainty was one of the many advantages of using the CNN-based method against the traditional peak-counting method. Although we described this point in Supplementary Text 2 of the original manuscript, we have further clarified it in the caption of Fig. 4 and in Supplementary Text 2.

Figures, line 373:

a) Compared to the traditional peak-counting method, the convolutional neural network (CNN)-based method produced the slope closer to 1 and higher R^2 . The uncertainty of individual estimations of the CNN-based method are color-coded in the right figure, while the traditional peak-counting method does not provide such uncertainty of estimation (Supplementary Text 2).

Supplementary information, line 176:

Our CNN was also designed to provide a measure of uncertainty in each prediction, allowing us to filter out high-uncertainty predictions in further analysis. In contrast, the traditional peak-counting method does not provide the uncertainty of its estimation; thus, we used a single color on the left figure of Fig. 4a. This information about uncertainty was one of the many advantages of using the CNN-based method against the traditional peak-counting method. Even without this uncertainty filtering, the CNN outperformed the traditional peak-counting method when tested on data unseen by the CNN during training (Fig. 4a; the slope is closer to 1, and R^2 is higher).

8. In Discussion section, it mentioned that the reason why the structural degenerations precede the flow counterparts might be the effect of a compensating adaptation. What's the specific compensation adaptation conjecture? Also, please give a possible explanation for the morphological parameters on the physiological functions, similar to that of topology parameters in this manuscript.

We have added related discussion to Supplementary Text 5.

Supplementary information, line 334, new paragraphs:

There are many models that explain the relationship between vascular structure and flow. In the most basic form commonly used in the literature, blood flow in a vessel can be modeled by Poiseuille's law^{4,26}. This postulates that flow is, $F \propto \frac{r^4 \Delta P}{\eta L}$ where r is the vessel radius, ΔP is the pressure difference along the vessel, η is the viscosity and L is the

vessel length. Flow is strongly related to the diameter of the vessel, with larger diameters increasing blood flow.

Based on this basic relation, one of the related compensation adaptations is likely the global autoregulation of cerebral blood flow (CBF). When the vessel radius decreases so that it tends to lower blood flow, the pressure can increase to maintain the blood flow at a similar level. This is a well-established principle of global autoregulation. Significant changes in CBF are not seen until this autoregulation is no longer able to compensate for the effect of structural degeneration, at which point flow begins to decrease²⁷.

Combining the morphology-flow model and the possible compensation adaptation, we can anticipate that since the compensating adaptation cannot be sustained indefinitely, eventually CBF would decrease due to the decreased vessel diameter. This explains one of our observations that blood flow decreased in penetrating vessels after the decrease in diameter.

Additionally, we have added further discussion to Supplementary Text 7, regarding how the observed morphological parameters can pose various effects on the physiological functions, especially when combined with the capillary flow result. The added discussion particularly focuses on two possible associations/consequences: hypertension and micro-hypoxia.

Supplementary information, line 396, new paragraph:

The combined results of arteriolar diameter/flow and capillary flow suggest another possible consequence of decreased arteriolar diameter and flow, namely an increase in the presence of hypoxic micro-pockets in the cortex. The observed decreases in both arteriolar diameter and flow would lead to arteriolar and near-arteriolar tissue oxygen pressure (pO₂)³⁴. This impaired arteriolar oxygen delivery would not immediately lead to an impairment in capillary-bed tissue pO₂ when the compensatory mechanism by enhanced capillary flow works as hypothesized above from our observed capillary RBC flux increases. However, when the compensatory mechanism no longer works at later ages, the capillary-bed tissue pO₂ would become lower and spatially more heterogeneous, as observed between 60 and 100 WOA in WT mice³⁴. This impaired capillary oxygen supply increases the presence of hypoxic micro-pockets, which may explain the observation of tinier microinfarcts in aging brains, particularly those with mild cognitive decline or AD^{35,36}. Therefore, the decrease in arteriolar diameter and flow observed in both WT and 3xTg mice with age may have important implications for the development and progression of neurodegenerative diseases.

The decrease in penetrating arteriole and venule diameters (Table 1) further suggests that the blood pressure might have increased with age in the 3xTg mice we used. Unfortunately, we did not longitudinally measure blood pressure in our study. Hypertension has been linked to AD, and animal studies suggest that hypertension can lead to amyloid plaques, neuroinflammation, blood-brain barrier dysfunction, and cognitive impairment³⁷. However, human studies on the association between hypertension and AD have been sparse and inconsistent, and the effect of

antihypertensive medications on AD seems weak^{38,39}. Investigating interactions among plaques, tangles, cerebrovascular pathology, and dementia may be key to understanding hypertension's role in AD development³⁹. Therefore, our future work includes performing a long-term trace of blood pressure using a noninvasive method like the tail-cuff technique in diverse AD models, which will enable us to compare the long-term blood pressure trace with the traces of cerebral microvascular properties obtained by the presented methods.

9. Please give the loss or accuracy curve about the CNN network used to predict RBC flux in supplementary information.

We have added the loss curve to Supplementary Fig. 6.

Supplementary information, line 66, updated figure:

Supplementary Figure 6. a) InceptionTime 1D CNN architecture for the prediction of RBC flux from OCT time series data. **b)** The loss curve during the training of the CNN.

10. Is it reasonable to use the TPM time traces to determine the ground truth? And when using it, how to solve the problem that the two-photon microscope will be used for long-term acquisition of red blood cell channel data? Is necessary to consider the possible problems of long-term acquisition.

We did not use TPM for long-term acquisition but used it only for training our CNN. In training the CNN, the “ground truth” data does not need to be from long-term acquisition. The CNN was only to measure the RBC flux from a given time trace, and later to be used in measuring the RBC flux from OCT time traces, where OCT data were acquired from different age points. As the reviewer is correctly concerned here, and as we argued in the Introduction section of the original manuscript, fluorescence TPM is not suitable for long-term longitudinal, repeated acquisition. We have further clarified this point in Supplementary Text 2.

Methods, line 537:

We trained a 1D CNN using the OCT time traces as the training data, and the corresponding RBC flux determined from the TPM time traces as ground truth (Supplementary Fig. 5 and Supplementary Text 2).

Supplementary information, line 187, new paragraph:

Regarding two-photon microscopy (TPM) data as the ground truth, TPM has been used as a standard method of measuring RBC flux and speed in capillary vessels of the rodent brain cortex⁸⁻¹³. It is based on fluorescence (staining plasma in most cases) and thereby yields a high signal-to-noise ratio. It is also capable of detecting RBC passage from vessels located deep in the cortical tissue, generally down to hundreds of micrometers deep, which cannot be achieved by using an older approach based on video microscopy¹⁴⁻¹⁶. Thus far, to the best of our knowledge, no techniques offer more accurate measurements of RBC flux and speed than TPM.

11. When training the CNN model, explain the ratio of the training set and the validation set.

The original manuscript already provided this information in the Methods section. We have slightly modified the sentence for further clarification.

Methods, line 537:

We augmented the training dataset by reversing the time of the time traces, and split this augmented dataset into training, validation and testing as 70%, 20% and 10% respectively.

12. Is it okay to calculate the mean curvature of the vessels using the mean derivative of the tangent along the vessel without considering the influence of the thickness of the blood vessel?

When measuring the tortuosity of a vessel, vessel thickness is not typically used as an input⁴. However, we acknowledge that tortuosity and thickness can be somewhat correlated, such as with thicker vessels being less tortuous. Therefore, one should be cautious when interpreting tortuosity results.

In our study, although the distributions were similar between AD and WT mice for both diameter and tortuosity (Fig. 3b), only the tortuosity exhibited a difference in the rate of change with age (the slope of LME fit) between the groups (Table 1). Hence, we interpret that the observed difference in the age-related change rate of tortuosity reflects true changes in vascular curvature rather than an influence of thickness changes. We have clarified this point in the manuscript.

Discussion, line 272, newly added:

The capillary tortuosity did not show significant difference in its fractional changes between AD and WT until the end of measurement (35 WOA), but its rate of change with age was positive in AD only. While vessel diameter and tortuosity can be correlated (e.g., with thicker vessels being less tortuous), it is unlikely that the observed significance in the tortuosity change rate of AD is an influence of diameter changes because the diameter change rate did not show significant difference between AD and WT (Table 1).

Methods, line 505:

We measured the capillary length and diameter as previously described²⁵, the histograms of which are shown in Fig. 3b. We also measured the capillary tortuosity from the mean curvature of the vessels, calculated by the mean derivative of the tangent along the vessel⁴². Since tortuosity and diameter can be correlated (e.g., with thick vessels being less tortuous), diameter results should be carefully considered when interpreting tortuosity results.

13. For each figure and result, the specific number of mice and the number of repetitions to measure the selected vessel should be given.

The original manuscript provided this information in the main text of the Result section, but we have added it to each figure caption as well.

Figures, line 335:

d) Time courses of the normalized vessel diameters (gray; 60 vessels from 7 AD mice, 47 vessels from 6 WT mice, 7 timepoints) and their LME fits (color).

Figures, line 346:

⁴ Bullitt E, Gerig G, Pizer SM, Lin W, Aylward SR. Measuring tortuosity of the intracerebral vasculature from MRA images. IEEE Trans Med Imaging. 2003 Sep;22(9):1163-71. doi: 10.1109/TMI.2003.816964. PMID: 12956271; PMCID: PMC2430603.

c) Time courses of arteriolar diameter of individual vessels (gray; 36 vessels from 7 AD mice, 29 vessels from 6 WT mice, 7 timepoints) and their LME fits (color). **d)** The LME fits of arteriolar diameter shown together. **e)** LME fits of venular diameter. **f)** Time courses of venular flow of individual vessels (gray; 47 vessels from 7 AD mice, 39 vessels from 6 WT mice, 7 timepoints) and their LME fits (color).

Figures, line 362:

c) Time courses of the normalized mean capillary length (gray; 7 AD mice and 6 WT mice, 7 timepoints) and their LME fits (color).

Figures, line 383:

e) RBC flux changes as tracked through the same animals (gray; 7 AD mice and 6 WT mice, 7 timepoints) and their LME fits.

Figures, line 390:

a) NOL test scores tracked by individual animals (gray lines; 7 AD mice and 6 WT mice, 9 timepoints) and their LME fits (color).

14. In line 516, the author defined points whose Cook's distance was greater than three times the mean value as outliers and eliminates them. Generally, the median is selected as the threshold.

Both median and mean are used in calculating Cook's distance. Examples of using the mean value include the MATLAB documentation⁵ and published papers^{6,7}. We chose the use of mean values because it tends to exclude less data points as outliers and thus is generally considered more conservative and inclusive. We have added the previous studies using the mean value to the Methods section.

Methods, line 595:

Outliers were excluded based on Cook's distance: any value with a Cook's distance greater than 3 times the mean was considered an outlier, as practiced in many studies^{48,49}.

⁵ <https://www.mathworks.com/help/stats/cooks-distance.html>

⁶ Brinkman L, Stolk A, Marshall TR, Esterer S, Sharp P, Dijkerman HC, de Lange FP, Toni I. Independent Causal Contributions of Alpha- and Beta-Band Oscillations during Movement Selection. *J Neurosci*. 2016 Aug 17;36(33):8726-33. doi: 10.1523/JNEUROSCI.0868-16.2016. PMID: 27535917; PMCID: PMC4987441.

⁷ Ramirez DG, Ciccaglione M, Upadhyay AK, Pham VT, Borden MA, Benninger RKP. Detecting insulinitis in type 1 diabetes with ultrasound phase-change contrast agents. *Proc Natl Acad Sci U S A*. 2021 Oct 12;118(41):e2022523118. doi: 10.1073/pnas.2022523118. PMID: 34607942; PMCID: PMC8521687.

15. In line 513, the author used the AIC value to select the optimal model, it should be further explained by combining the ANOVA analysis results.

We did not perform ANOVA in this study. Repeated measures ANOVA (rANOVA) is another method that could be used in analyzing longitudinal data, but we used linear mixed effects (LME) models as clarified in the Methods section of the original manuscript. This is because rANOVA is known to be vulnerable to effects from missing values, imputation, inequivalent time points between subjects, and violations of sphericity⁸, which can result in sampling bias and inflated rates of Type I error⁹. LME analysis is often preferred over rANOVA when analyzing longitudinal data¹⁰.

Regarding the use of AIC values, we fitted both linear and logistic functions to data to figure out which trend better explains the data. In this course, we chose the better-fitting model by comparing the AIC scores of the two fits. This AIC-based selection is widely used (e.g., references added to the revision as shown below). We have added these citations to the related part.

Methods, line 591:

..., and then selected the model with the lowest Akaike Information Criterion (AIC) following the widely used method^{46,47}.

16. In line 529, the author determined the age at which AD and WT groups differed by assessing the degree of overlap between two confidence intervals. How to quantitatively measure the degree of overlap?

We simply measured the degree of overlap as follows: First, we computed the distance between the lower bound of the confidence interval for the group with higher values and the upper bound of the confidence interval for the other group. Next, we selected the age at which this distance became larger than zero as the age at which the AD and WT groups started to differ.

This analysis aimed to determine at which time point the two groups could be considered statistically different using only the confidence intervals of the fitted lines. For example, if the 95% confidence intervals of two variables just touch, what is the approximate p-value of the t-test between these variables? One might expect this value to be $p = 0.05$, but it is not true. A simple numerical investigation of the relationship between t-tests and confidence intervals showed that the p-value would be approximately 0.005 with the sample size of our study. The question then is: which percentage confidence intervals would result in a p-value of $p = 0.05$ when the two confidence intervals touch? This

⁸ Gueorguieva R, Krystal JH. Move over ANOVA: progress in analyzing repeated-measures data and its reflection in papers published in the Archives of General Psychiatry. Arch Gen Psychiatry. 2004 Mar;61(3):310-7. doi: 10.1001/archpsyc.61.3.310. PMID: 14993119.

⁹ Keith E. Muller & Curtis N. Barton (1989) Approximate Power for Repeated-Measures ANOVA Lacking Sphericity, Journal of the American Statistical Association, 84:406, 549-555, DOI: 10.1080/01621459.1989.10478802

¹⁰ Krueger C, Tian L. A comparison of the general linear mixed model and repeated measures ANOVA using a dataset with multiple missing data points. Biol Res Nurs. 2004 Oct;6(2):151-7. doi: 10.1177/1099800404267682. PMID: 15388912.

question is a well-studied one, suggesting that the variables are statistically different at the point at which the 93% confidence intervals touch (after multiple-hypothesis correction).

We have added these details to the related part as follows.

Methods, line 611:

For these metrics, we determined the age at which AD and WT groups differed by assessing the degree of overlap between two confidence intervals (CIs). To account for multiple hypothesis testing using Bonferroni correction, we aimed to find the age at which the groups were significantly different at a rate of $\alpha = 0.05/7$, which in turn leads to the age at which the 93% CIs just touch. In detail, we computed the distance between the lower CI bound of the group with higher values and the upper CI bound of the other group. The age from which this distance becomes larger than zero was selected as the age at which AD and WT groups started to differ.

REVIEWERS' COMMENTS

Reviewer #1 (Remarks to the Author):

The authors responded well to my comments and performed additional experiments to address some of them, thanks. Appropriate changes were made in the manuscript. Nonetheless, I have two additional minor remarks.

From my comment of the original manuscript:

-It would be quite interesting to compare the results on OCT microangiography obtained here with a more global technique, e.g. perfusion MRI or transcranial Doppler ultrasound. On the other hand, for validation purposes it would have been important to compare the OCT results with histological analysis of microvasculature, e.g. using collagen IV and transglutaminase 2 (see e.g. Hohsfield LA et al., Mol Cell Neurosci 2014; 63:83-95, doi: 10.1016/j.mcn.2014.10.006).

The authors provided references from hypoperfusion in early human AD. In addition to these, they should also provide references to hypoperfusion in animal models, especially if available for the 3xTg animal model.

Additional comment to the revised manuscript:

-More details about the age and genetic background of the mice especially of the 3xTg animals along with an appropriate reference about the transgenic model should be provided in the Methods section.

Reviewer #2 (Remarks to the Author):

In tracking neuronal and microvascular changes, a very important issue should also be considered. Recent studies have reported the coupling effect between neuronal activities and extracellular space, as well as with the vascular compartment. The following studies and results should be discussed in the discussion section.

(1. Li Y, et al. The Mechanism of Downregulated Interstitial Fluid Drainage Following Neuronal Excitation. Aging Dis. 2020 Dec 1;11(6):1407-1422. doi: 10.14336/AD.2020.0224. PMID: 33269097; PMCID: PMC7673848.

2. Wang A, et, al. The Drainage of Interstitial Fluid in the Deep Brain is Controlled by the Integrity of Myelination. Aging Dis. 2019 Oct 1;10(5):937-948.

3. Wang R, et, al. The Alteration of Brain Interstitial Fluid Drainage with Myelination Development. Aging Dis. 2021 Oct 1;12(7):1729-1740

4. Gao Y, et, al. Early changes to the extracellular space in the hippocampus under simulated microgravity conditions. Sci China Life Sci. 2022 Mar;65(3):604-617)

And since the authors have responded to all comments, I suggest accepting this paper for publication.

Response to Review

Manuscript ID: NCOMMS-22-09196A

Reviewer 1

The authors responded well to my comments and performed additional experiments to address some of them, thanks. Appropriate changes were made in the manuscript. Nonetheless, I have two additional minor remarks.

We thank the reviewer for reading our responses and revisions, as well as for suggesting the additional improvements.

From my comment of the original manuscript:

-It would be quite interesting to compare the results on OCT microangiography obtained here with a more global technique, e.g. perfusion MRI or transcranial Doppler ultrasound. On the other hand, for validation purposes it would have been important to compare the OCT results with histological analysis of microvasculature, e.g. using collagen IV and transglutaminase 2 (see e.g. Hohsfield LA et al., Mol Cell Neurosci 2014; 63:83-95, doi: 10.1016/j.mcn.2014.10.006).

The authors provided references from hypoperfusion in early human AD. In addition to these, they should also provide references to hypoperfusion in animal models, especially if available for the 3xTg animal model.

We have added studies that reported hypoperfusion from AD model mice.

Discussion, line 266:

Although the primary focus of our study is technical demonstration rather than biological discovery (Supplementary Text 4), the observed cortical hypoperfusion in AD is

consistent with findings from perfusion magnetic resonance imaging and transcranial Doppler ultrasound studies, which have reported hypoperfusion in various cortical regions in early human AD (see Refs. 33,34 for reviews), as well as findings from AD model mice [Niwa 2000, Niwa 2002].

Additional comment to the revised manuscript:

-More details about the age and genetic background of the mice especially of the 3xTg animals along with an appropriate reference about the transgenic model should be provided in the Methods section.

We have added the information to the Methods section.

Methods, line 330:

Male wild-type (WT, C57BL/6J, Jackson Lab) mice (n = 10) and male 3xTg AD mice (B6;129-Tg(APP^{Swe},tau^{P301L})1Lfa Psen1^{TM1Mpm}/Mmjax, Jackson Lab, n = 10) underwent craniotomy at 10 weeks of age (WOA) for window placement³⁸ and in vivo OCT imaging under inhaled anesthesia.

Reviewer 2

In tracking neuronal and microvascular changes, a very important issue should also be considered. Recent studies have reported the coupling effect between neuronal activities and extracellular space, as well as with the vascular compartment. The following studies and results should be discussed in the discussion section.

1. Li Y, et al. The Mechanism of Downregulated Interstitial Fluid Drainage Following Neuronal Excitation. *Aging Dis.* 2020 Dec 1;11(6):1407-1422. doi: 10.14336/AD.2020.0224. PMID: 33269097; PMCID: PMC7673848.
2. Wang A, et, al. The Drainage of Interstitial Fluid in the Deep Brain is Controlled by the Integrity of Myelination. *Aging Dis.* 2019 Oct 1;10(5):937-948.
3. Wang R, et, al. The Alteration of Brain Interstitial Fluid Drainage with Myelination Development. *Aging Dis.* 2021 Oct 1;12(7):1729-1740
4. Gao Y, et, al. Early changes to the extracellular space in the hippocampus under simulated microgravity conditions. *Sci China Life Sci.* 2022 Mar;65(3):604-617)

And since the authors have responded to all comments, I suggest accepting this paper for publication.

We thank the reviewer for reading our responses and revisions, as well as for suggesting the additional improvement. We have added a new paragraph to the Discussion section.

Discussion, line 306:

A β clearance plays an important role in AD [Tarasoff-Conway 2015]. Various overlapping and interacting clearance mechanisms are being studied, including the glymphatic pathway. One of these clearance systems, called perivascular system, involves a number of components such as the cerebrospinal fluid, interstitial fluid, periaxonal space, water channel aquaporin-4 in astrocytes [Iliff 2012], extracellular space affected by astrocytic endfeet structure [Li 2020], and fiber myelination [Wang 2019, Wang 2021]. Since many of these components are structurally and functionally interrelated with blood vessels, the methods presented in this paper would be useful in studying how the clearance system components interact with the related vascular properties, how the interaction varies with age, and how the age-dependent interaction differs between AD and normal aging.